# Causally Reliable Concept Bottleneck Models

**Giovanni De Felice**[*]
Università della Svizzera Italiana
giovanni.de.felice@usi.ch

**Arianna Casanova Flores**[*]
University of Liechtenstein

**Francesco De Santis**[*]
Politecnico di Torino

**Silvia Santini**
Università della Svizzera Italiana

**Johannes Schneider**
University of Liechtenstein

**Pietro Barbiero**[†]
IBM Research

**Alberto Termine**[†]
Scuola Universitaria Professionale della Svizzera Italiana, IDSIA

## Abstract

Concept-based models are an emerging paradigm in deep learning that constrains the inference process to operate through human-interpretable variables, facilitating explainability and human interaction. However, these architectures, on par with popular opaque neural models, fail to account for the true causal mechanisms underlying the target phenomena represented in the data. This hampers their ability to support causal reasoning tasks, limits out-of-distribution generalization, and hinders the implementation of fairness constraints. To overcome these issues, we propose *Causally reliable Concept Bottleneck Models* (C$^2$BMs), a class of concept-based architectures that enforce reasoning through a bottleneck of concepts structured according to a model of the real-world causal mechanisms. We also introduce a pipeline to automatically learn this structure from observational data and *unstructured* background knowledge (e.g., scientific literature). Experimental evidence suggests that C$^2$BMs are more interpretable, causally reliable, and improve responsiveness to interventions w.r.t. standard opaque and concept-based models, while maintaining their accuracy.

## 1  Introduction

In recent years, interpretable neural models have become more popular, achieving performance similar to powerful opaque Deep Neural Networks (DNNs) (Alvarez Melis & Jaakkola, 2018; Chen et al., 2019, 2020). Among these, Concept Bottleneck Models (CBMs) (Koh et al., 2020; Zarlenga et al., 2022; Yuksekgonul et al., 2022; Barbiero et al., 2023) guarantee high expressivity and interpretability by enforcing DNNs to reason through a layer of high-level, human-interpretable variables called *concepts* (e.g., the "color" and "shape" of an object) (Kim et al., 2018; Achtibat et al., 2023; Fel et al., 2023). In CBMs, a neural encoder first maps the raw input to concepts, forming a semantically transparent intermediate representation that is used by a simple decoder for downstream predictions. Beyond transparency, this design allows human experts to intervene on mispredicted concepts at test time to improve downstream task predictions (Espinosa Zarlenga et al., 2024).

However, like standard DNN architectures, CBMs remain pure *associative* models (Pearl, 2019): their decision-making process reflects statistical correlations within the data rather than real-world causal mechanisms. As a result, they fail to distinguish between spurious correlations and true causal

---

[*]Equal contribution.
[†]Equal senior authors.

39th Conference on Neural Information Processing Systems (NeurIPS 2025).

relationships. Recognizing this distinction is fundamental to achieving a *reliable* scientific understanding, supporting causal reasoning for intervention (Pearl, 2009; Peters et al., 2017), enabling *robust* generalization under distributional shifts, and the implementation of fairness constraints (Schölkopf et al., 2021; Wang et al., 2022).

To address these limitations, we propose *Causally reliable Concept Bottleneck Models* (C²BMs): a class of concept-based architectures that enforce reasoning through a "Causal Bottleneck" (Fig. 1) of concepts structured according to a model of the real-world causal mechanisms underlying data generation. C²BMs process information as follows. First, a neural encoder extracts a set of latent representations from raw data. Then, information flows from latent representations through a given causal graph where each node represents an interpretable variable (e.g., "smoker", "bronchitis"). At inference time, the value of each variable is predicted from its causal parents through an interpretable structural equation, parametrized adaptively by a hypernetwork.

Designing a C²BM requires identifying domain-relevant concepts and specifying their causal relationships, a process that depends heavily on expert knowledge, which could be scarce, costly, or entirely unavailable in practice. To mitigate this reliance and favor agile deployment across domains, we propose a fully automated pipeline (Fig. 1, Causal Graph Construction) for instantiating a C²BM, in which the set of relevant concepts and the causal graph are automatically learned from a mixture of data and *unstructured* background knowledge.

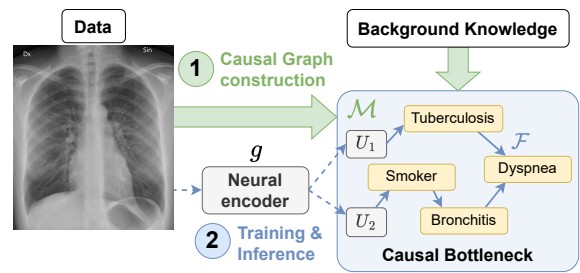

Figure 1: *Causally reliable Concept Bottleneck Models* (C²BMs) enforce reasoning through a "Causal Bottleneck" aligned with a model of real-world causal mechanisms obtained from data and background knowledge.

Experimental evidence shows that C²BMs: (i) improve on **consistency** with real-world causal mechanisms, **without compromising accuracy** w.r.t. standard DNN models, CBMs, and their extensions (Sec. 5.1); (ii) **improve interventional accuracy** on downstream concepts with fewer interventions (Sec. 5.2); (iii) mitigate reliance on spurious correlations (**debiasing**, Sec. 5.3); (iv) permit interventions to remove unethical model behavior and **meet fairness requirements** (Sec. 5.4).

## 2    Preliminaries

We introduce the notation and key formalizations underlying standard CBMs and causal modeling. A more detailed background on causality is provided in App. A.

**Concept Bottleneck Models.**    CBMs (Koh et al., 2020) are interpretable-by-design architectures that explain their predictions using high-level interpretable variables called *concepts*. Standard CBMs decompose prediction into two stages: a neural encoder maps the input $X$ to a set of intermediate concepts $\mathcal{V} = \{V_i\}_{i=1}^C$, and a decoder predicts the target $Y$ from $\mathcal{V}$. This yields:

$$P(Y, \mathcal{V} \mid X) = \underbrace{P(Y \mid \mathcal{V})}_{\text{decoder}} \ \underbrace{P(\mathcal{V} \mid X)}_{\text{concept encoder}} . \tag{1}$$

Concept Embedding Models (CEMs) (Zarlenga et al., 2022) enhance CBMs by pairing concepts with high-dimensional embeddings of the form $P(\mathcal{U} \mid \mathcal{V}, X)$, where $\mathcal{U} = \{U_i\}_{i=1}^C$. These embeddings are provided to the decoder to predict the target variable $Y$, enabling the model to achieve performance comparable to standard DNN approaches while maintaining semantic interpretability. Critically, traditional decoders rely on a *bipartite structure* assumption, wherein all concepts are treated as direct causes of the target, e.g., $Y = f(V_1, \ldots, V_C)$ for CBMs. This assumption is often overly simplistic for real-world problems. Bringing the reasoning of concept-based architectures closer to real-world mechanisms constitutes the main focus of this work.

**Causal Reliability.**    A model $\mathcal{M}$ is *causally reliable* w.r.t. a target phenomenon $T$ if and only if the structure of $\mathcal{M}$'s decision-making process is consistent with the causal mechanisms underlying $T$

(Termine & Primiero, 2024). Although state-of-the-art DNNs and concept-based models offer high expressivity, they lack causal reliability.

**Structural Causal Models.** The standard framework for modeling causal mechanisms is the *structural causal model* (SCM) (Bareinboim et al., 2022). An SCM $\mathcal{M}$ is a tuple $\langle \mathcal{V}, \mathcal{U}, \mathcal{F}, P \rangle$, where:

- $\mathcal{V}$ is a set of $C$ *endogenous* variables, modeling observable magnitudes of interest;
- $\mathcal{U}$ is a set of *exogenous* variables, modeling unobservable magnitudes determined by factors external to $\mathcal{V}$;
- $\mathcal{F} = \{f_i\}_{i=1}^{C}$ is a set of functions such that

$$V_i = f_i(\mathsf{PA}_i, \mathcal{U}_i) \quad \forall i = 1, \ldots, C \tag{2}$$

  where $\mathsf{PA}_i \subseteq \mathcal{V} \setminus V_i$ is the set of the *endogenous* parents of $V_i$, $\mathcal{U}_i \subseteq \mathcal{U}$ is an exogenous parent summarizing all the information influencing $V_i$ that is not explicitly represented in $\mathcal{V}$, and the entire set $\mathcal{F}$ forms a mapping from $\mathcal{U}$ to $\mathcal{V}$.
- $P(\mathcal{U})$ is a joint probability distribution over $\mathcal{U}$.

Each SCM can be associated with a graphical representation in which nodes correspond to the variables, and edges encode the functional relationships specified by $\mathcal{F}$. Here, we focus on SCMs whose associated graph is a *directed acyclic graph* (DAG) (Pearl, 1995; Zaffalon et al., 2020b). In most cases, the underlying DAG is unknown and must be inferred from observational data, a process known as *causal discovery* (Peters et al., 2017; Zanga et al., 2022). However, methods based solely on observational data cannot generally guarantee the identification of a unique DAG (Peters et al., 2017). The set of candidate DAGs can be refined by incorporating additional information, which we refer to as *background knowledge* (Andrews et al., 2020; Abdulaal et al., 2023). This can be drawn from a range of sources, such as human experts, structured repositories of information (e.g., domain ontologies), or "unstructured" samples of information (e.g., scientific papers or other documentation).

## 3 Related works

Traditional concept-based architectures impose a strict bipartite structure in which concept neuron activations are assumed to directly cause task outputs (Koh et al., 2020; Yuksekgonul et al., 2022; Kim et al., 2023; Oikarinen et al., 2023; Yang et al., 2023; Barbiero et al., 2023; Vandenhirtz et al., 2024). This strong, often unrealistic assumption can lead to misleading explanations. For example, attributing a lung cancer diagnosis to both a 'cough' and 'smoker' concept could risk the false interpretation that reducing coughing could reduce cancer risk. Moreover, most CBMs assume independence among concepts, which is unrealistic, as it ignores natural co-occurrences (e.g., 'smoke' and 'fire') and prevents improvements in one concept from propagating to related concepts during interventions. Stochastic CBM (SCBM) (Vandenhirtz et al., 2024) and Concept Graph Models (Dominici et al., 2025) attempt to relax this assumption. However, these approaches capture only associations rather than causal relations, making it vulnerable to spurious correlations in the data. To date, no methodology exists for structuring the concept bottleneck according to a reliable causal model.

Recent approaches like DiConStruct (Moreira et al., 2024), aim to improve this aspect by generating causal graphs linking concepts to opaque DNN predictions. However, DiConStruct is a *post-hoc* method that may misalign with the original DNN's outputs and relies solely on observational data, neglecting background knowledge and resulting in under-determined causal structures. Other architectures, such as Neural Causal Models (Ke et al., 2019) and Neural Causal Abstractions (Xia & Bareinboim, 2024), impose even stronger assumptions, requiring access to either the true causal graph or a low-resolution structural causal model, which are impractical in many cases.

## 4 Method

In this section, we introduce *Causally reliable Concept Bottleneck Models* ($C^2$BMs) and the pipeline we propose to fully automate its instantiation, learning, and functioning (Fig. 2).

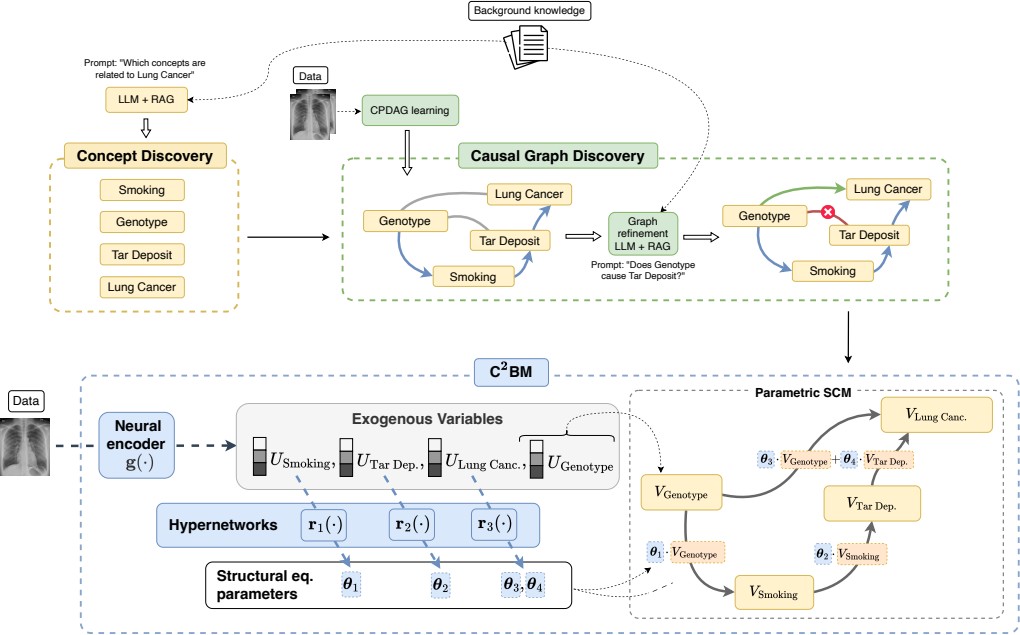

Figure 2: **Overview of the C²BM fully automated pipeline.** The pipeline consists of three key blocks: (i) Concept discovery: discovery and labeling of the relevant variables $\mathcal{V}$ from background knowledge; (ii) Causal graph discovery: discovery of the causal graph by integrating data and background knowledge; (iii) the C²BM model, comprising a neural encoder and an adaptively parametrized SCM. Once the model is trained, it can support forward and interventional queries about any endogenous variable (e.g., predicting *dyspea*).

## 4.1 Causally reliable Concept Bottleneck Models

A C²BM is a concept-based architecture that leverages the formalism of SCMs to structure a "causal bottleneck of concepts". More formally, let: (i) $X$ denoting a random variable modeling (possibly noisy) input features; (ii) $\mathcal{V} = \{V_i\}_{i=1}^{C}$ be a set of $C$ semantically meaningful variables modeled as *endogenous* variables; (iii) $\mathbf{G}$ be a DAG connecting variables in $\mathcal{V}$. A C²BM is a neural architecture implementing the tuple $\langle \mathbf{g}, \mathcal{M}_{\boldsymbol{\Theta}} \rangle$ where:

- $\mathbf{g}(\cdot)$ is a *neural encoder* modeling a probability distribution $P(\mathcal{U}|X)$ over a set of latent, high-dimensional embeddings $\mathcal{U} = \{U_i\}_{i=1}^{C}$, representing the *exogenous* variables;

- $\mathcal{M}_{\boldsymbol{\Theta}}$ is a *parametric* SCM $\langle \mathcal{V}, \mathcal{U}, \mathcal{F}_{\boldsymbol{\Theta}}, P(\mathcal{U}|X) \rangle$ (see Sec. 2), where we assume a parametric form for the functions' set. Specifically, the structure of the functions is determined by the connectivity of $\mathbf{G}$, and the parameters $\boldsymbol{\Theta}$ are predicted from $\mathcal{U}$ by a hypernetwork.

The information flowing along a C²BM can be described as follows (Fig. 2, right side). First, the values of the exogenous variables $\mathcal{U}$ are predicted using the exogenous encoder $\mathbf{g}(\cdot)$ from $X$. Then, the information flows along the SCM $\mathcal{M}_{\boldsymbol{\Theta}}$ starting from the endogenous sources (predicted from $\mathcal{U}$) down to the sinks. At each subsequent level of the causal graph, the values of each $V_i$ are predicted from the values of its *parents* $\mathsf{PA}_i$ based on the relative structural equation $f_i \in \mathcal{F}_{\boldsymbol{\Theta}}$.

## 4.2 Model instantiation

To instantiate a C²BM, one requires a labeled dataset $\mathcal{D}$ annotated for all variables in $\mathcal{V}$, as well as a DAG capturing the causal relationships among $\mathcal{V}$. However, such resources may be inaccessible, problem-specific, or heavily dependent on human expertise. To address this challenge, we propose a fully automated pipeline that enables the use of C²BM also in such complex scenarios. Our approach extracts the necessary components from: (i) a potentially unlabeled dataset $\mathcal{D}_x$; (ii) a potentially unstructured repository of background knowledge $\mathcal{K}$.

Our pipeline (see Fig. 2) addresses the following sub-problems: (i) causal graph construction, which includes concept discovery, concept labeling (Sec. 4.2.1), and causal graph discovery (Sec. 4.2.2); and (ii) training of the neural parameters of the encoder and the hypernetwork determining the structural equations (Sec. 4.3). In the following, we outline our implementation for each sub-problem. Specifically, building $C^2BM$'s individual prerequisites will be mostly based on prior work. Note that integrating them into a coherent, automated pipeline is instead part of this paper's contributions.

*Remark* 4.1. Concepts and causal graph constitute an input for $C^2BM$, which could remain agnostic to how they are obtained, e.g., provided by human experts. Notably, alternative or novel approaches may be employed, provided they solve the same problems (Loula et al., 2025).

### 4.2.1 Concept discovery and labeling

**Problem 4.2** (Concept Discovery). *Given a dataset of i.i.d. samples $\mathcal{D}_x = \{\mathbf{x}_i\}_{i=1}^{N}$, and a background knowledge repository $\mathcal{K}$ relative to a task, identify a set of relevant variables $\mathcal{V}$.*

In the CBM community, automated concept discovery and labeling using Large Language Models (LLMs) has become a standard solution when human supervision is unavailable (Oikarinen et al., 2023; Yang et al., 2023; Srivastava et al., 2024; Yamaguchi & Nishida, 2025). In our implementation, we follow the label-free CBM approach from Oikarinen et al. (2023), where concepts are discovered by querying an LLM for those most relevant to the task. We then apply a filtering procedure to retain only concepts that meet criteria such as brevity, distinctiveness (i.e., not too similar to each other or the target), and presence in the training data.

Once $\mathcal{V}$ are selected, we label the dataset with variable annotations $\mathcal{D} = \{(\mathbf{x}_i, \mathbf{v}_i)\}_{i=1}^{N}$ to supervise concept learning. To do so, we adopt a strategy similar to Oikarinen et al. (2023), leveraging a pre-trained contrastive vision-language model, such as CLIP (Radford et al., 2021). This projects both data samples and the discovered concept names into a shared embedding space and computes their alignment to generate concept labels. Full implementation details are provided in App. B.

### 4.2.2 Causal graph discovery

**Problem 4.3** (Causal Discovery). *Let $\mathbf{G}^*$ be the true, unknown, graph over a set of variables $\mathcal{V}$ from which a dataset $\mathcal{D}$ was generated. The causal discovery problem consists in recovering $\mathbf{G}^*$ from the observed dataset $\mathcal{D}$ (Zanga et al., 2022).*

As anticipated in Sec. 2, a promising direction for addressing this problem is to combine standard causal discovery algorithms with knowledge-base querying. In our pipeline, we focus on a well-known class of methods that recover an equivalence class of graphs from data, referred to as the *Markov equivalence class* (MEC) (Spirtes et al., 2001; Pearl, 2009; Zanga et al., 2022).This class can be compactly represented as a *Completed Partially Directed Acyclic Graph* (CPDAG), an extension of a DAG where edges remain unoriented when there is insufficient evidence in the data to infer causal direction. This is a desirable property as it prevents incorrect (spurious) orientations based on the data alone. Specifically, we apply the *Greedy Equivalence Search* (GES) (Chickering, 2002) algorithm, which we found performed well empirically (see App. G.3). Then, we leverage a pre-trained LLM to assess each undirected edge in the CPDAG, orienting the ones corresponding to true causal relationships while discarding those originating from spurious correlations. To improve the robustness and generalizability of this approach, we pair the LLM with a *Retrieval Augmented Generation* (RAG) technique, which is known to reduce hallucinations and can provide problem-specific knowledge. To further improve robustness and performance, we repeat each query 10 times, selecting the most frequent outcome (Wang et al., 2023). Implementation details are provided in App. C.

### 4.3 Structural equations and model training

**Problem 4.4** (Learning structural equations). *Let $\mathcal{E}$ be the set of edges describing causal connections in a DAG $\mathbf{G}$ connecting variables in $\mathcal{V}$. Given $\mathcal{D} = \{(\mathbf{x}_i, \mathbf{v}_i)\}_{i=1}^{N}$ and $\mathcal{E}$, predict the parameters $\mathbf{\Theta}$ of the structural equations $\mathcal{F}_{\mathbf{\Theta}}$.*

We model the structural functions $f_i \in \mathcal{F}_\Theta$ describing the causal mechanisms relating each endogenous variable [3] to its endogenous parents as weighted linear sums, i.e., for each $i$:

$$V_i = \sum_{V_j \in \mathsf{PA}_i} [\boldsymbol{\theta}_{f_i}]_j V_j \tag{3}$$

where $\mathsf{PA}_i$ denotes the set of endogenous parents of $V_i$ [4]. For the parameters $\boldsymbol{\theta}_{f_i}$, we do not learn a single parameterization; instead, these are adaptively inferred for each different realization of $X$, by a *hypernetwork* $\mathbf{r}(\cdot)$ (Ha et al., 2017; Barbiero et al., 2023; Debot et al., 2024), based on the values of the exogenous variables $\mathcal{U}$ and the graph connectivity. In our implementation, we consider separate hypernetworks $\mathbf{r}_i(\cdot)$ (e.g., independent DNNs), each taking as input a separate exogenous variable:

$$\boldsymbol{\theta}_{f_i} = \mathbf{r}(\mathcal{E}, \mathcal{U})_i := \mathbf{r}_i(U_i) = \mathbf{r}_i(\mathbf{g}(X)_i). \tag{4}$$

The design in Eq. 3 and 4 improves both interpretability and expressivity. To aid (mechanistic) interpretability, structural equations take a linear form (Eq. 3): the value of each is a weighted linear combination of its parents. The adaptive re-parameterization of the equation's weights performed by the hypernetwork $\mathbf{r}(\cdot)$ allows the model to also approximate non-linear relationships among endogenous variables (see App. D, which also includes a proof that $C^2BM$ is a universal approximator, regardless of the underlying causal graph). This idea is in line with existing literature on interpretability (Ribeiro et al., 2016; Alvarez Melis & Jaakkola, 2018) and concept-based methods (Barbiero et al., 2023) [5]

**Model training.**    The training of $C^2BM$ consists of learning the neural parameters of the encoder $\mathbf{g}(\cdot)$ and hypernetwork $\mathbf{r}(\cdot)$ *end-to-end* from the input data. We formalize this by modeling the joint conditional distribution $P(\mathcal{V}, \mathcal{U}, \boldsymbol{\Theta} \mid X, \mathcal{E})$ which factorizes as:

$$P(\mathcal{V}, \mathcal{U}, \boldsymbol{\Theta} \mid X, \mathcal{E}) = \overbrace{P(\mathcal{V} \mid \mathcal{U}; \boldsymbol{\Theta})}^{\text{endogenous}} \overbrace{P(\boldsymbol{\Theta} \mid \mathcal{E}, \mathcal{U})}^{\text{structural equation}} \overbrace{P(\mathcal{U} \mid X)}^{\text{exogenous}} \tag{5}$$

Figure 3: Probabilistic graphical model of $C^2BM$ inference.

where: $P(\mathcal{U} \mid X)$ represents the exogenous encoder $\mathbf{g}(\cdot)$; $P(\boldsymbol{\Theta} \mid \mathcal{E}, \mathcal{U})$ represents the hypernetwork $\mathbf{r}(\cdot)$ predicting the structural equations' parameters using the given causal connections and the exogenous variables; $P(\mathcal{V} \mid \mathcal{U}; \boldsymbol{\Theta})$ represents a causally reliable classifier leveraging the structural equations $\mathcal{F}_\Theta$ to predict the values of the endogenous variables. Under the Markov condition imposed by the $C^2BM$ causal graph, the causally-reliable classifier can be re-written as a product of independent distributions i.e.,

$$P(\mathcal{V} \mid \mathcal{U}; \boldsymbol{\Theta}) = \prod_i P(V_i \mid \mathsf{PA}_i, U_i; \mathbf{r}(\mathcal{E}, \mathcal{U})_i) \tag{6}$$

where $U_i = \mathbf{g}(X)_i$ and $\mathbf{r}(\mathcal{E}, \mathcal{U})_i = \boldsymbol{\theta}_{f_i}$. From the above factorization, we derive the $C^2BM$'s training objective, which corresponds to maximizing the empirical log-likelihood of the training data:

$$\phi^* = \arg\max_\phi \sum_{\mathcal{D}} \sum_{i=1}^{C} \log P(V_i \mid \mathsf{PA}_i, U_i; \mathbf{r}(\mathcal{E}, \mathcal{U})_i) \tag{7}$$

*Remark* 4.5. We clarify that we do not claim to identify the true structural functions. Instead, $C^2BM$ numerically approximates the outcomes as if they were generated by the underlying (unknown) structural equations. This approximation, along with $C^2BM$'s DAG, is sufficient to compute reliable interventions, which is a core objective in the concept-based community (Poeta et al., 2023; Steinmann et al., 2024).

---

[3]Root variables are predicted from the exogenous variables $\mathcal{U}$ using a neural network. Further details are provided in App. D.

[4]We refer to $V_i$ as a variable to allow for a more general formalization. In a classification setting, concepts assume categorical values; hence $V_i$ represents the probability of a concept state activation.

[5]This idea also aligns with Balke & Pearl (1994) and Zaffalon et al. (2020a), where exogenous variables are used to represent relationships between endogenous variables when the structural equations are unknown.

Table 1: Task accuracy (%). Task concepts are as follows: dysp (Asia), Akt (Sachs), PropCost (Insurance), BP (Alarm), R5Fcst (Hailfinder), parity (cMNIST), mouth slightly open (CelebA), Survival (CUB$_C$), pneumothorax (Pneumoth.). $^*$ refers to reduced concept bottlenecks. Methods matching the performance of OpaQNN and showing a significant improvement over the other considered methods are highlighted in bold. Uncertainties represent 2 sample mean $\sigma$ across 5 runs.

| MODEL | SEMANTIC TRANSP. | CAUSAL REL. | ASIA | SACHS | INSURANCE | ALARM | HAILFINDER | cMNIST | CELEBA | CUB$_C$ | PNEUMOTH. | ASIA$^*$ | ALARM$^*$ |
|---|---|---|---|---|---|---|---|---|---|---|---|---|---|
| OPAQNN | ✗ | ✗ | $71.0_{\pm1.4}$ | $65.83_{\pm.71}$ | $66.8_{\pm1.5}$ | $62.8_{\pm1.5}$ | $72.0_{\pm1.9}$ | $91.24_{\pm.72}$ | $74.97_{\pm.08}$ | $60.3_{\pm0.9}$ | $80.0_{\pm1.5}$ | $71.0_{\pm1.4}$ | $62.8_{\pm1.5}$ |
| CBM$_{+lin}$ | ✔ | ✗ | $71.2_{\pm1.6}$ | $65.44_{\pm.93}$ | $67.1_{\pm1.7}$ | $62.7_{\pm1.3}$ | $72.2_{\pm2.3}$ | $93.92_{\pm.37}$ | $71.07_{\pm.48}$ | $56.8_{\pm4.2}$ | $76.6_{\pm0.8}$ | $56.0_{\pm8.3}$ | $52.7_{\pm1.0}$ |
| CBM$_{+mlp}$ | ✔ | ✗ | $71.2_{\pm1.4}$ | $65.68_{\pm.84}$ | $66.7_{\pm1.4}$ | $62.3_{\pm1.8}$ | $70.9_{\pm2.2}$ | $93.55_{\pm.31}$ | $71.27_{\pm.20}$ | $56.2_{\pm1.9}$ | $76.7_{\pm0.6}$ | $58.6_{\pm2.7}$ | $52.8_{\pm1.2}$ |
| CEM | ✔ | ✗ | $71.1_{\pm1.8}$ | $65.93_{\pm.72}$ | $66.7_{\pm1.6}$ | $60.8_{\pm1.1}$ | $71.5_{\pm1.9}$ | $93.72_{\pm.26}$ | $\mathbf{74.72}_{\pm.14}$ | $56.2_{\pm2.0}$ | $\mathbf{80.1}_{\pm1.1}$ | $\mathbf{69.7}_{\pm2.0}$ | $\mathbf{61.8}_{\pm1.2}$ |
| SCBM | ✔ | ✗ | $70.7_{\pm1.6}$ | $66.30_{\pm.55}$ | $67.1_{\pm1.7}$ | $63.4_{\pm1.5}$ | $73.4_{\pm2.1}$ | $94.02_{\pm.23}$ | $72.15_{\pm.15}$ | $59.9_{\pm1.4}$ | $78.4_{\pm0.6}$ | $61.8_{\pm1.2}$ | $53.5_{\pm0.9}$ |
| **C²BM** | ✔ | ✔ | $71.4_{\pm1.7}$ | $65.33_{\pm1.1}$ | $66.4_{\pm1.5}$ | $62.5_{\pm1.4}$ | $74.1_{\pm1.8}$ | $94.18_{\pm.03}$ | $\mathbf{74.73}_{\pm.41}$ | $58.1_{\pm1.1}$ | $\mathbf{80.5}_{\pm0.7}$ | $\mathbf{70.8}_{\pm1.7}$ | $\mathbf{60.5}_{\pm1.4}$ |

## 5 Experimental evaluations

We evaluate the performance of the proposed C²BM pipeline. Experiments are conducted across different datasets and settings, allowing for the investigation of the following aspects: classification accuracy (Sec. 5.1), causal reliability (Sec. 5.1), accuracy under ground-truth interventions (Sec. 5.2), debiasing (Sec. 5.3), and fairness (Sec. 5.4). App. G provides additional results and ablations.

The considered datasets include both synthetic and real-world benchmarks. As synthetic datasets, we sample $10^4$ points from each of the five following discrete Bayesian networks available from the `bnlearn` repository (Scutari, 2010): **Asia** (Lauritzen & Spiegelhalter, 1988), **Sachs** (Sachs et al., 2005), **Insurance** (Binder et al., 1997), **Alarm** (Beinlich et al., 1989), and **Hailfinder** (Abramson et al., 1996). We include **cMNIST**, a variant of the original dataset (LeCun et al., 2010) in which the image data are colored according to custom rules. Additionally, we consider three real-world datasets: **CelebA** (Liu et al., 2015), a facial recognition dataset labeled with different binary facial attributes; **CUB$_C$**, a custom version of the original bird image dataset (He & Peng, 2019) from which we select a subset of concepts and define new ones to introduce deeper causal relationships; **Siim-Pneumothorax** (You et al., 2023), containing chest X-ray images annotated with a single label indicating the presence of pneumothorax, without additional concepts or their annotations. To generate them, we follow the label-free approach outlined in Sec. 4.2.1. Exhaustive details on all datasets are given in App. E.

The performance of the proposed pipeline is investigated alongside an opaque neural baseline predicting the task variable only (**OpaqNN**) and established state-of-the-art (SOTA) concept-based architectures, namely: **CBM** (Koh et al., 2020), with linear and non-linear decoder; **CEM** (Zarlenga et al., 2022); and **SCBM** (Vandenhirtz et al., 2024). Hyperparameters have been selected via an independent search for each dataset–model pair based on performance on the validation set. Further details on each model's architecture and hyperparameters can be found in App. F. Note that all baselines, except for OpaqNN, provide concept-based explanations for their predictions and allow concept interventions at test-time. This excludes architectures such as Self-Explainable Neural Networks (Alvarez Melis & Jaakkola, 2018) and Concept Whitening (Chen et al., 2020) as they do not offer a clear mechanism for intervening on their concept bottlenecks. We also excluded other CBM baselines such as Probabilistic CBMs (Kim et al., 2023), Post-hoc CBMs (Yuksekgonul et al., 2022), Label-free CBMs (Oikarinen et al., 2023; Yang et al., 2023), as all of them share the same limitation of vanilla CBMs and CEMs: the causal graph is fixed and bipartite. Python code for reproducing all experiments is provided alongside the submission as supplementary material.

### 5.1 Task accuracy and causal reliability

Our initial experiment evaluates task accuracy. For each dataset, we designate a predefined single variable as the prediction *task*. All models except OpaqNN are trained to predict the task while simultaneously learning to fit the remaining concepts. Tab. 1 presents the task accuracy for all evaluated models (see App.G.1 for concept accuracy). To further assess model expressiveness, we also evaluate task accuracy on modified versions of the *Asia* and *Alarm* datasets, where selected concepts (App. E) are intentionally removed to create a stronger bottleneck.

Table 2: Structural Hamming distance (App. F.1) and number of mistaken edges between true and learned DAG. Reliability of standard flat CBMs is reported for reference. The total number of edges is in parentheses.

| METRIC | AFTER | CMNIST | ASIA | SACHS | INSUR. | ALARM | HAILF. |
|---|---|---|---|---|---|---|---|
| HAMMING | FLAT CBM | 1.0 | 6.5 | 11.75 | 36.5 | 45.0 | 69.0 |
| | CD | 0.2 | 0.7 | 3.4 | 6.4 | 5.4 | **11.0** |
| | CD + LLM | **0** | **0.3** | **1.8** | **6.3** | **5.0** | **11.0** |
| INCORRECT | FLAT CBM | 1 (1) | 11 (8) | 23 (17) | 74 (52) | 78 (46) | 117 (66) |
| EDGES | CD | 1 (1) | 3 (8) | 17 (17) | 19 (52) | 13 (46) | **22** (66) |
| (TRUE EDGES) | CD + LLM | **0** (1) | **1** (8) | **7** (17) | **18** (52) | **10** (46) | **22** (66) |

**$C^2$BM achieves comparable or higher accuracy to non-causally reliable models (Tab. 1).** Our evaluation shows that $C^2$BM achieves robust accuracy across datasets, matching the performance of the expressive models OpaqueNN and CEM. Notably, as the concept bottleneck is reduced, $C^2$BM retains expressivity by leveraging exogenous variables to propagate residual information from the input. This is in contrast with CBMs implementing a hard bottleneck.

**$C^2$BM improves on causal reliability (Tab. 2).** $C^2$BM captures a rich causal structure that aligns well with real-world dependencies. We quantitatively assess this alignment by comparing the learned and true causal graphs in synthetic datasets. Tab. 2 reports two metrics: a structural Hamming distance (detailed in App. F.1) and the number of incorrect edges, computed after causal discovery (CD) and refinement via LLM queries. Metrics for the simplistic graphs from CBMs (all concepts are treated as mutually independent and direct causes of the task) are reported for reference. Results indicate that integrating CD with background knowledge produces a causal graph that is more accurately aligned with the true structure. Notably, on the *Sachs* dataset,

the integration of background knowledge enables to correctly identify 10 additional edges w.r.t. CD alone. Detailed ablation studies on causal graph discovery methods and LLM types are provided in App. G.3-G.4-G.5. To further validate the quality of the learned causal graph, App. G.2 demonstrates that $C^2$BM achieves comparable task accuracy using either the learned or the true graph. When considered together, the results in Tab. 1-2 highlight $C^2$BM's ability to improve on causal reliability without compromising expressivity and performance.

### 5.2 Ground-truth interventions

After training all models on the same classification task as in Sec. 5.1, we test their responsiveness to ground-truth interventions, i.e., replacing predicted concepts with ground-truth values [6]. This simulates a form of human intervention in a deployed model. Following each intervention, we compute the average accuracy over all variables (concepts and task) prediction. As for the policy, we intervene on random concepts within progressively deeper levels in the hierarchy defined by the true graph. This constitutes the only intervention policy aligned with real-world causal-effect relationships. When the true graph is unavailable, we use the one generated by our pipeline.

**$C^2$BM improves accuracy on downstream concepts with fewer interventions (Fig. 4).** Our findings, reported in Fig. 4, demonstrate that $C^2$BM achieves higher accuracy improvements with fewer interventions compared to alternative models. This advantage stems from two key properties of $C^2$BM: (i) unlike other baselines that do not account for connections among concepts, interventions on an upstream concept in $C^2$BM directly influence **all** downstream nodes, potentially enhancing the predictions of their values; (ii) unlike SCBM, the effects of interventions in $C^2$BM are restricted to concepts that are causally related, rather than altering concept values due to spurious correlations.

---

[6]Ground-truth interventions can be seen as a special case of causal *do*-interventions (see App. A.1), where variables are set to their ground-truth values.

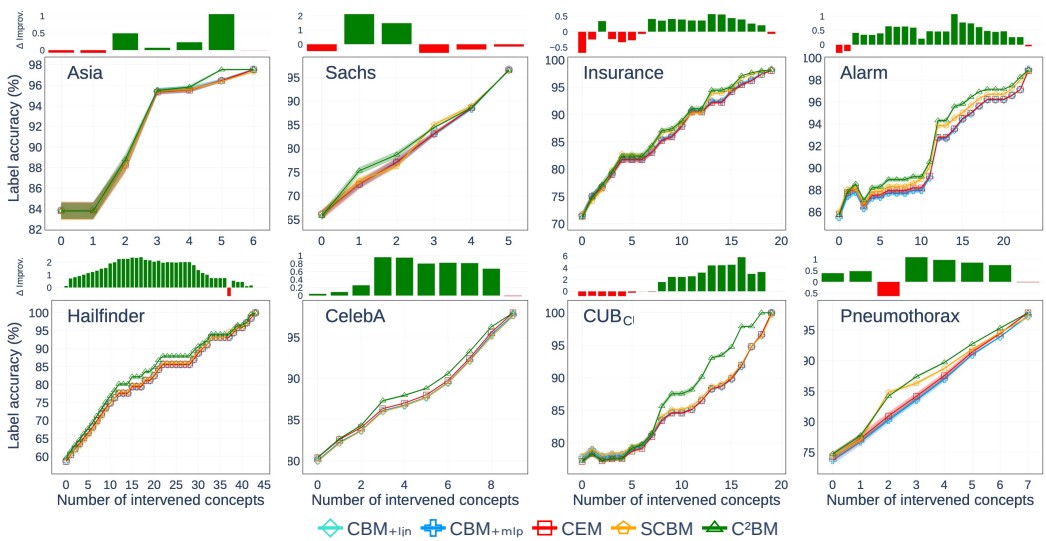

Figure 4: Label accuracy (%) on downstream variables (task included) after intervening on concepts up to progressively deeper levels in the graph hierarchy. Summit plots show the difference of $C^2BM$'s accuracy w.r.t. the best-performing baseline. Uncertainties represent 2 sample mean $\sigma$ across 5 runs.

## 5.3 Debiasing

We hypothesize that a real-world-aligned causal bottleneck can reduce reliance on spurious correlations. To test this, we use *cMNIST*, where digit *Color* is correlated with *Parity* during training (all the odd digits are green). At test time, the correlation among *Color* and *Parity* is reversed (all the even digits are green), introducing a distribution shift that challenges generalization. The neural encoders in all models still struggle with out-of-distribution (OOD) generalization. Therefore, as expected, all models including $C^2BM$, fail to extrapolate correctly, capturing the artificial correlation. However, their enforced reasoning is different. All baselines retain the concept-task connections, perpetuating the color-parity shortcut. In contrast, $C^2BM$ detects the color-parity edge through causal discovery but correctly removes it via graph refinement with the LLM block. This is reflected in large differences in performance after concept interventions, which alleviate (or even remove) the impact of the encoder.

**Causal bottlenecks mitigate reliance on spurious correlations (Fig. 5).** Fig. 5 shows the accuracy on *Parity* after ground-truth interventions on each concept. As expected, color has no effect across all models, confirming the learned bias. Notably, $C^2BM$ exhibits the largest improvement when intervening on the *number* concept (achieving $\sim 90\%$ accuracy), as its causal structure isolates color and strengthens training on the correct feature. Although a comprehensive analysis of OOD robustness is beyond the scope of this paper, our results suggest that the $C^2BM$ pipeline holds promise in improving generalization in biased settings.

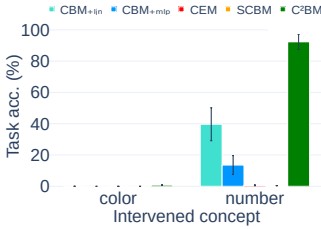

Figure 5: Biased ColorMNIST dataset. Task accuracy on *Parity* after ground-truth interventions.

## 5.4 Fairness

We create a customized *CelebA* dataset to evaluate the influence of sensitive attributes on the decision-making of the model. Specifically, we consider a hypothetical scenario in which an actor with a specific physical attribute is required for a specific role. However, the hiring manager has a strong bias toward *Attractive* applicants. To model this, we define two custom attributes: *Qualified*, indicating whether an applicant meets the biased hiring criteria, and *Should be Hired*, which depends on both

*Qualified* and the task-specific requirement (*Pointy Nose*). In our fairness analysis, we aim to intervene and remove any such unfair bias.

**$C^2$BMs permit interventions to meet fairness requirements (Tab. 3).** We measure the Causal Concept Effect (CaCE) (Goyal et al., 2019) of *Attractive* on *Should be Hired* before and after blocking the only path between them, i.e., performing an intervention on *Qualified*. Tab. 3 shows that $C^2$BM is the only model able to successfully remove the influence, achieving a post-intervention CaCE of 0.0%. This difference stems from the model architectures: in CBM, CEM, and SCBM, all concepts are directly connected to the task, meaning interventions on one concept cannot block the influence of others. In contrast, $C^2$BM enforces a structured causal bottleneck allowing for interventions to fully override the effects of parent nodes and block information propagation through the intervened node. This highlights $C^2$BM's ability to enforce causal fairness by eliminating biased pathways.

Table 3: CelebA dataset. Causal Concept Effect (CaCE, %) between a sensitive concept (*Attractive*) and a target concept (*Should be hired*), before and after blocking the path between the two variables with do-interventions.

| CACE METRIC | $CBM_{+lin}$ | $CBM_{+mlp}$ | CEM | SCBM | $C^2$BM |
|---|---|---|---|---|---|
| BEFORE INT. | $12.3_{\pm 1.1}$ | $12.5_{\pm 3.1}$ | $19.0_{\pm 3.6}$ | $30.9_{\pm 4.1}$ | $25.1_{\pm 1.8}$ |
| AFTER INT. | $21.9_{\pm 1.3}$ | $11.8_{\pm 6.5}$ | $8.2_{\pm 2.2}$ | $14.8_{\pm 3.6}$ | $\mathbf{0.0}_{\pm 0.0}$ |

## 6 Conclusions

We presented $C^2$BM, a concept-based model advancing prior research by structuring the bottleneck of concepts according to a model of causal relations between human-interpretable variables. By combining observational data with background knowledge, $C^2$BMs improves on causal reliability without compromising performance. This offers several additional benefits, e.g., improved interventional accuracy, robustness to spurious correlations, and fairness.

**Applications and broader impact.** We speculate $C^2$BM has the potential to significantly narrow the hypothesis space in complex scientific domains where even a panel of human experts might struggle to identify or exclude plausible hypotheses worth testing. For instance, constructing the hypothesis space to design clinical trials accounting for the influence of environmental conditions on gastrointestinal biochemistry requires deep interdisciplinary knowledge—not only in environmental science and biochemistry, but also in genetics, microbiology, nutrition science, and epidemiology, among others. In such settings, $C^2$BM may integrate human scientific knowledge across these diverse fields to construct a comprehensive causal graph, thereby supporting experts in systematically excluding hypotheses that are inconsistent with the integrated body of evidence. This can help accelerate interdisciplinary scientific discovery, while reducing experts' cognitive burden.

**Limitations.** $C^2$BM requires a robust prior knowledge base, access to pre-trained models (LLMs for causal discovery), and well-crafted prompts for querying these LLMs. Biases within the knowledge base or in the observational data can reduce the system's reliability (though $C^2$BM still outperforms the selected baselines). Furthermore, the SOTA in causal structural learning currently faces scalability limitations, which also constrain $C^2$BM. As these techniques become more scalable and robust, $C^2$BM stands to benefit. Finally, encoder embeddings used to construct exogenous variables can move out of distribution, and since the encoder's OOD performance is not guaranteed, the SCM's OOD performance may likewise be affected.

**Future works.** Future directions include a deeper investigation of OOD generalization with $C^2$BMs, an extensive exploration of their role in causal inference (e.g, *counterfactual queries* extending (), see App. A), and the identification of optimal intervention policies. Moreover, incorporating PAGs (Zhang, 2008) would enable modeling of hidden confounders.

## Acknowledgments

This work is supported by the Swiss National Science Foundation (SNSF) through the grant 205121_197242 for the project "PROSELF: Semi-automated Self-Tracking Systems to Improve Personal Productivity" and the Hasler Foundation under the Project ID: 2024-05-15-70. AC and JS acknowledge support from FFF of the University of Liechtenstein grant lbs_24_08. PB acknowledges support from the Swiss National Science Foundation Postdoctoral Fellowships IMAGINE (No. 224226) and has received funding from the Research Foundation Flanders (FWO, G033625N). AT acknowledges the support by the Hasler Foundation grant Malescamo (No. 22050), and the Horizon Europe grant Automotif (No. 101147693).

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

# Appendix

## Table of Contents

## A   Extended background on causality

### A.1   Pearl's framework of causality

Contemporary research in causal inference and causal machine learning predominantly builds on the framework introduced by Pearl (2009) (see also Pearl (2019)). This framework centers on an agent's ability to reason about underlying causal mechanisms, going beyond mere statistical associations

observed in data. Pearl formalizes this capacity through the notion of answering different types of *what-if* questions, structured into a three-level hierarchy (see Figure 6).

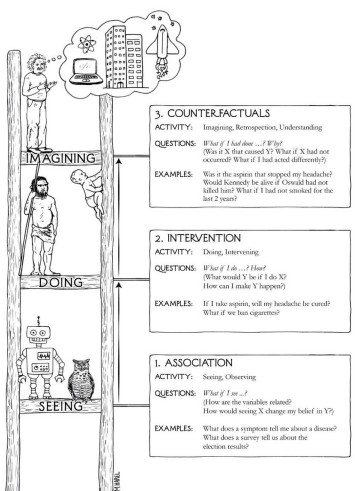

Figure 6: Pearl's Hierarchy as represented in Pearl & Mackenzie (2018).

- **Level 1: Observational questions** ("what is the value of $Y$ if I *observe* $X = x$?"). These questions rely purely on statistical associations within the data. They can be answered using standard tools such as conditional probabilities or expectations, with no need for causal assumptions.

- **Level 2: Interventional questions** ("what is the value of $Y$ if I *do* $X = x$?"). These questions focus on the effects of actively intervening on variables, rather than passively observing them. The expression $do(X = x)$ denotes such an intervention, where $X$ is externally set to $x$, overriding its natural causes.

  To compute such an intervention, one typically relies on a graphical model representing causal dependencies among variables, e.g., a directed acyclic graph (DAG). The intervention is then modeled by removing all incoming edges to $X$, effectively simulating the setting of $X$ independently of its original causes. This modified graph is then used to compute the post-intervention distribution $P(Y \mid do(X = x))$.

- **Level 3: Counterfactual questions** ("What would have been the value of $Y$ if I had observed $X = x'$ instead of $X = x$?"). These questions focus on states of affairs alternative to the actual reality. They constitute the upper layer of Pearl's hierarchy and their computation requires a detailed knowledge of the causal mechanisms relating the variables of the target-problem.

While standard machine learning models [7] can generally handle only observational questions, reasoning at interventional and counterfactual levels necessitates dedicated causal models and inference methods (Pearl, 2009; Peters et al., 2017).

## A.2 Causal opacity

**Problem A.1** (Causal Opacity). *Given a DNN model $\mathcal{M}$ and an user $A$, we say that $\mathcal{M}$ is* causally-opaque *with respect to $A$ whenever $A$ is not capable to understand the inner causal structure of $\mathcal{M}$'s decision-making process. The opposite of causal opacity is* causal transparency.

Note that **causal transparency does not presuppose or imply causal reliability**, the two issues, although related, are indeed very different. Consider for example a concept-based model $\mathcal{M}_1$, where the graph representing how the concepts are connected to the final task is shown in the left panel

---

[7]This excludes models specifically designed for causal inference, such as those in the field of *causal machine learning* (Kaddour et al., 2022).

of the figure below (a). This model is trained to predict whether a given colored image represents an even or odd number (variable $P$, for "parity") passing through a bottleneck of two interpretable concepts, namely "number" ($N$) and "color" ($C$). The structure of $\mathcal{M}_1$'s decision-making process is causally-transparent, including two edges connecting the final task $P$ with $N$ and $C$ respectively. However, this causal structure is not consistent with the causal structure of the world (b), for which $C$ and $P$ are clearly independent concepts Therefore, $\mathcal{M}_1$ cannot be considered *causally reliable*, although it is *causally transparent*.

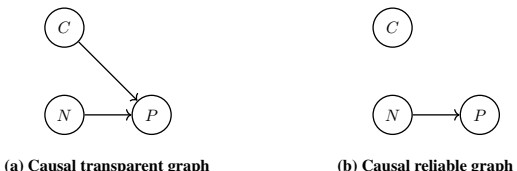

(a) Causal transparent graph          (b) Causal reliable graph

Causal opacity partially depends on another opacity issue of central relevance for our analysis, i.e., *semantic opacity*.

**Problem A.2** (Semantic Opacity). *Given a model $\mathcal{M}$ and a user $A$, we say that $\mathcal{M}$ is* semantically opaque *to $A$ if and only if $\mathcal{M}$'s decision-making process is based on features that do not possess any interpretable meaning for $A$.*

Semantic opacity is addressed by concept-based architectures, such as concept-bottleneck models and their extensions, which we discuss in the main paper.

# B    Concept discovery details

CBMs-like architectures tipically rely on labeled data for each concept, a costly process that can hinder their practical adoption. Label-free Concept Bottleneck Models (Label-free CBMs) (Oikarinen et al., 2023) address this issue by automatically generating concepts and assign concept labels using pre-trained models.

In our paper, we apply an approach similar to the one followed in Oikarinen et al. (2023) to the Siim-Pneumothorax dataset, which lacks concepts and concept annotations. Using GPT-4o, we first generate candidate concepts with a specific prompt (see App. C.3), and apply a multi-stage filtering process:

- discard too long concepts (>50 character);
- filter out concepts too similar to class labels or to each other. Specifically, we use the CXR-CLIP model – a CLIP-based model pretrained on medical imaging datasets (Johnson et al., 2019; Irvin et al., 2019; Wang et al., 2017; You et al., 2023) – to encode both concepts and class labels, and discard any concept with cosine similarity > 0.9 to either a class label or another concept;
- discard concepts that are not sufficiently present in the training data. To this end, we also compute image embeddings using CXR-CLIP, and remove concepts whose maximum cosine similarity with images in the training set is < 0.2.

We then annotate images by computing their similarity to each remaining concept using CXR-CLIP embeddings, and binarize the resulting scores via 2-means clustering.

# C    Causal graph discovery details

## C.1    Causal discovery algorithms

Algorithms for addressing the causal discovery problem can be broadly classified into two main categories (Peters et al., 2017):[8]

---

[8]Here, we restrict our discussion to causal discovery algorithms that assume the set of observed variables sufficiently captures the relevant causal influences. For more complex scenarios, see, e.g., (Peters et al., 2017).

- **Independence-based methods**. These methods assume a correspondence between conditional independence in the data and graphical separation among variables, leveraging this relationship to infer the underlying graph structure. Typically, they recover a class of DAGs that are equivalent with respect to conditional independencies, which can be compactly represented by a CPDAG.

- **Score-based methods.** These methods define a scoring function over potential graph structures and search for the graph that maximizes it, often using criteria such as the *Bayesian Information Criterion* (BIC) (Peters et al., 2017). The results from score-based methods are often comparable to those from independence-based approaches, as graphs that violate conditional independencies tend to result in poor model fits.

For our experiments, we adopted the *Greedy Equivalence Search* (GES) algorithm (Chickering, 2002), a score-based method that performed well in our setting, as shown in Table 8 in App. G.3.

GES is a two-phase greedy algorithm that searches over equivalence classes of DAGs (CPDAGs). It begins with the empty graph and iteratively adds edges that yield the greatest improvement in the scoring function. Once a local optimum is reached — where no addition improves the score — the algorithm enters a backward phase, greedily removing edges that most increase the score. The core idea is to navigate the space of CPDAGs through local transformations (edge additions and deletions), using the score as a guide to optimize structure learning.

In our implementation, we use the GES algorithm provided by the `causal-learn` Python library (Zheng et al., 2024), employing the *Bayesian Dirichlet equivalent uniform* (BDeu) scoring criterion for discrete variables (Heckerman et al., 1995; Chickering, 2002).

## C.2  LLM and RAG

LLMs (Brown et al., 2020; Jiang et al., 2024) are capable of answering complex queries without additional training. However, they can be unreliable and prone to hallucinations (Ji et al., 2023; Zhang et al., 2023). Retrieval Augmented Generation (RAG) (Lewis et al., 2020) addresses this by retrieving relevant textual information and appending it to the query, thereby improving the accuracy and reliability of LLM outputs. In the context of structural learning, LLMs have been utilized to construct causal graphs by employing ad-hoc prompts specifically designed to condition the model for answering causal queries (Antonucci et al., 2023; Long et al., 2023; Zhang et al., 2024). To enhance the causal graph discovery process, we utilize an LLM integrated with a RAG to either direct or eliminate edges that remain undirected by the causal discovery algorithm. The information retrieval process is outlined as follows:

1. *Document Retrieval*: Given a causal query (e.g., "Is lung cancer influenced by smoking?"), we first retrieve a set of documents from the web. In our experiments, we employed both the DuckDuckGo search engine for web pages and Arxiv for relevant paper abstracts. From each source, we retrieve the top 10 documents. For certain datasets (cMNIST, CelebA, and Sachs), local documents were used due to the unavailability or inaccessibility of information online (e.g., the Sachs paper). Each retrieved document is then segmented into smaller pieces, referred to as chunks, using a sliding window approach that samples a 512-token chunk every 128 tokens.

2. *Ranking*: The causal query is transformed by the LLM using a *query transformation* approach (Gao et al., 2023), which improves the semantic alignment of the query with the relevant document chunks. After transformation, all the retrieved chunks and the modified causal query are processed by a sentence transformer (Reimers, 2019), specifically the `multi-qa-mpnet-base-dot-v1` model. At this stage, cosine similarity is computed between the embedded transformed causal query and each embedded chunk.

3. *Context*: The LLM is then tasked with answering the causal query using the additional context retrieved in the previous steps. This context is derived from the top 5 chunks with the highest cosine similarity to the transformed causal query, providing the LLM with relevant and supportive information for generating a more accurate answer.

All the prompts mentioned in this section can be found in App. C.3.

## C.3 Prompts

In this subsection we list all the prompts used in both the causal discovery part and label free concept generation.

DUCKDUCKGO SEARCH PROMPT

```
Your task is to create the most effective search query to find information that answers
the user's question.
Your query will be used to search the web using a web engine (e.g. google, duckduckgo).
NOTE: be short and concise.

This is the question: {question}.
Provide the final query without brackets.
```

ARXIV SEARCH PROMPT

```
Your task is to create the most effective search query to find information that answers
the user's question.
Your query will be used to search scientific articles from the web.
From the given query, produce a query that will help to find the most relevant articles.
NOTE: be short and concise.

This is the question: {question}.
Provide the final query without brackets.
```

TRANSFORMATION QUERY PROMPT

```
Rephrase the query to align semantically with similar target texts while maintaining
its core meaning.
Output the expanded query enclosed within
<expanded_query> tags (e.g. <expanded_query>[example_query]</expanded_query>).
NOTE: be very short and concise.

Query: {query}
Expanded Query:
```

CAUSAL PROMPT

```
You are an expert in causal inference and logical analysis.
I will provide you with two concepts and you have to infer the causal relationship between them.
**Concept 1:** {concept_1} - {concept_1_description}
**Concept 2:** {concept_2} - {concept_2_description}

Now, use your knowledge and, if available, the context provided, to determine
which of the following options is the correct one:
(A) changing {concept_1} to certain values result in a change in {concept_2};
(B) changing {concept_2} to certain values result in a change in {concept_1};
(C) there is no causal relationship or reciprocal influence between {concept_1} and {concept_2}.

The following information are extracted from recent and reliable sources:
{context}

The answer has to be enclosed within <answer> tags (e.g. <answer>A</answer>).
Analyze the situation step-by-step to ensure the final conclusion is accurate.
```

CONCEPTS GENERATION PROMPT

```
You are an expert of {context}.
You need to list the most important features to recognize {class_label} from {input}.
List also the variables that are most likely to be associated with {class_label} as well as
the variables that are most likely to be associated with the absence of {class_label}.
You need also to give a list of superclasses for the word {class_label}.
Combine all the lists in a single one and separate the single terms with a comma.
If a term is composed by more than one word, use an underscore to separate the words.
```

# D  C$^2$BMs detailed architecture

In this appendix, we provide a detailed description of the proposed C$^2$BM model training and functioning, using the *Asia* dataset as an illustrative example and *Dyspnea* as the task (Fig. 7). We assume the following information is available:

- A set of human-understandable variables relevant to determining the task's value. Specifically, the binary concepts: {*Smoker, Bronchitis, Lung cancer, Either, Tubercolisis, Been in Asia, Xray anomalies, and Dyspnea*}.

- A training dataset $D = \{\mathbf{x}_i, \mathbf{v}_i\}_{i=1}^n$, where each sample is annotated with the values of all endogenous variables, i.e., the target variable *Dyspnea* and all preceding binary concepts.

- A DAG $\mathbf{G}$ outlining the causal relationships between the concepts and the task.

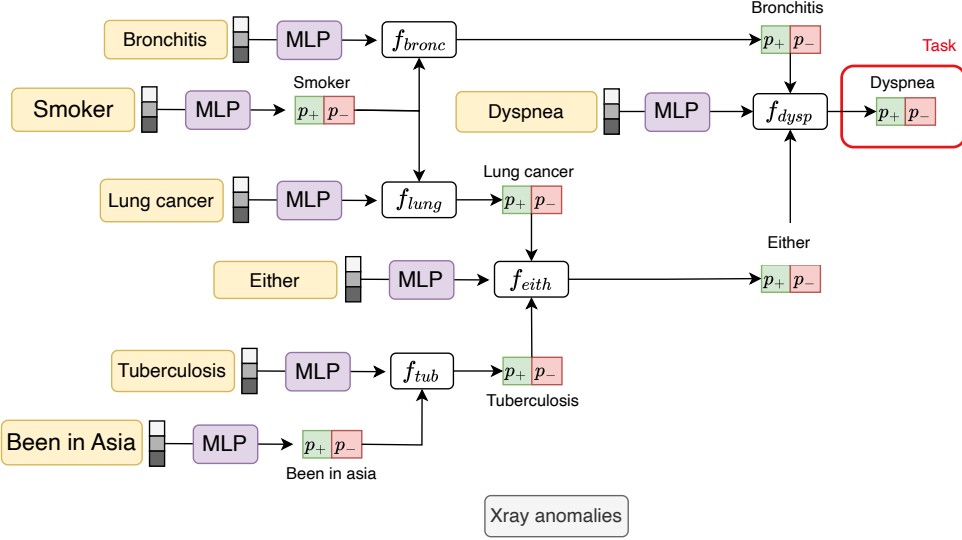

Figure 7: Detailed C$^2$BM architecture applied to the Asia dataset.

These elements can be either provided by human experts or generated using the pipeline we propose in the paper, as described in Fig. 2. The proposed approach follows four main steps (Fig. 7), as detailed below:

1. **Sub-causal Graph Selection.** We extract from the DAG $\mathbf{G}$ only the variables that are ancestors of the task, that is, variables for which there exists a causal path in $\mathbf{G}$ connecting the variable node to the task. In the case of the Asia dataset, the variable *X-ray anomalies* is discarded because it is not an ancestor of the task.

2. **Exogenous embeddings.** Each variable (including the task) is assumed to have an associated latent factor, which is represented as an embedding (the grey encoder symbols in Fig. 7) learned from the input using a dedicated neural encoder. In our implementation, these are implemented as MLPs, preceded by a dataset-specific feature extractor, e.g., a CNN for image data (as detailed in the App. E).

3. **Structural Equation Modeling.** We model the structural relationships between parent nodes and their child nodes using linear equations. The weights of these equations are predicted by separate hypernetworks, implemented as MLPs, which take as input the embedding of the child node produced in the previous step. Applying these functions we can derive the normalized logits of a child node from the ones of its parents. For the nodes that lie in the roots of the causal graph (*Been in Asia*, *Smoker*), their logits are obtained directly from the corresponding exogenous embeddings.

   For instance, we can calculate normalized logits for *Dyspnea* as follows:

$$\mathbf{p}_{Dyspnea} = \sigma(\boldsymbol{\theta}_1 \mathbf{p}_{Bronchitis} + \boldsymbol{\theta}_2 \mathbf{p}_{Either}) \tag{8}$$

   where $\mathbf{p}_{Bronchitis}$ and $\mathbf{p}_{Either}$ are the normalized logits for the parents, $\boldsymbol{\theta}_{f_{Dyspnea}} = [\boldsymbol{\theta}_1, \boldsymbol{\theta}_2]$ their corresponding weights and $\sigma$ denotes a transformation function (such as a softmax) applied to the weighted sum of the parent node logits, ensuring the final output is in a suitable range.

While the structural equations are linear, the fact that the weights can be adaptively inferred from exogenous variables allows us to capture complex dependencies between variables. This idea is analogous to locally approximating complex (smooth) functions. For example, consider an exponential relationship between two endogenous variables, $V_2 = e^{V_1}$ (Fig. 8). This function can be locally approximated by a linear form $V_2 = \theta_1 V_1$, where the weight $\theta_1$ is adjusted based on the value of $V_1$.

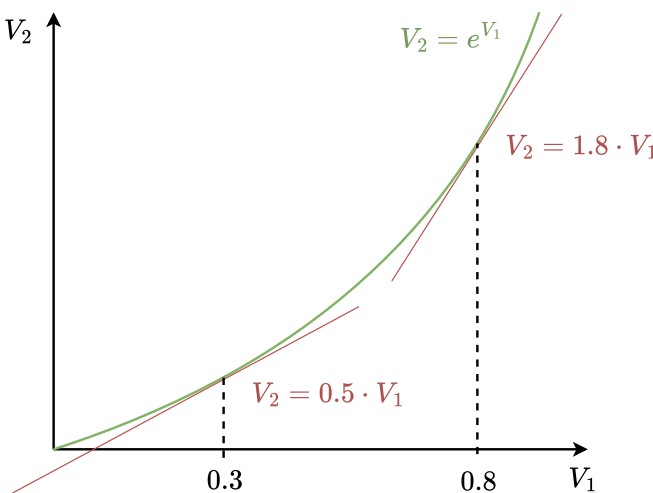

Figure 8: Non-linear functions can be modeled by adaptive re-parametrizable linear models.

It is important to notice that such a model supports queries about any specific endogenous variable. Specifically, after training, C$^2$BM can be used to predict a target variable (task), while using the other variables as concepts to explain the reasoning process. This provides greater flexibility compared to other concept-based architectures, which instead assume a fixed task. Notably, only the task's ancestors are relevant in this case and, in our implementation, all other concepts are discarded.

### D.1  C$^2$BM as an universal approximator

We establish the following result regarding the expressivity of C$^2$BM.

**Theorem D.1.** *C$^2$BM is a universal approximator regardless of the underlying causal graph.*

*Proof.* We show that C$^2$BM can predict any endogenous variable as any DNN. Assume the exogenous encoder $\mathbf{g}(\cdot)$ is a DNN (or any universal approximator) and that endogenous variables are represented as logits (the extension to fuzzy or Boolean values is trivial). We consider four cases:

1. **No endogenous parent:** If $V_i$ is a root, $V_i = \mathrm{MLP}_i(\mathbf{g}(X)_i)$. Since both $\mathbf{g}(\cdot)$ and $\mathrm{MLP}_i$ are universal approximators, their composition is also a universal approximator.

2. **Single endogenous root parent:** If $V_i$ has exactly one root parent $V_j$, then $V_i = [\boldsymbol{\theta}_{f_i}]_j V_j = [\mathbf{r}_i(\mathbf{g}(X)_i)]_j \cdot V_j$ and since both $\mathbf{r}_i(\cdot)$ and $\mathbf{g}(\cdot)$ are universal approximators, their composition is also a universal approximator. Multiplying this by $V_j$, which itself is produced by a universal approximator, preserves the ability to approximate any function of the input.

3. **Multiple endogenous root parents:** If $V_i$ has more than one parent, $\boldsymbol{\theta}_{f_i}$ can assign zero weights to all but one parent, reducing the case to a single-parent scenario.

4. **Non-root endogenous parents.** The reasoning above can be applied recursively following the topological ordering of the causal graph. Each variable is computed as a function of its parents, and universal approximation is preserved layer by layer.

Hence, by recursively composing universal approximators along the causal graph, C²BM can approximate any mapping from the input to endogenous variables. This holds for any graph structure, establishing C²BM as a universal approximator.

□

## D.2 C²BM interpretability

In this section, we present an explanation generated by C²BM on the *Asia* dataset. As shown in Tab. 2, the causal graph retrieved by the causal discovery mechanism is almost equal to the real one, despite a missing edge between *Been in Asia* and *Tuberculosis*. Starting from the source endogenous variables, it is possible to see the weight associated to each descending endogenous variable and the corresponding activation probability. For instance, *Lung cancer* is 'True' because the corresponding probability is peaked toward it ($P(1) = 0.99$). In particular, the decision-making process for the classification of *Dyspnea* as 'False' is completely unveiled. Although the parameter on the edge from *Bronchitis* to *Dyspnea* promotes a positive prediction, the stronger, negatively weighted connection from *Either* to *Dyspnea* dominates. Resulting in *Dyspnea* being predicted as 'False'. It is worth noting that the endogenous variable *X-ray anomalies* is not considered by C²BM's inference since it is not an ancestor of the defined task.

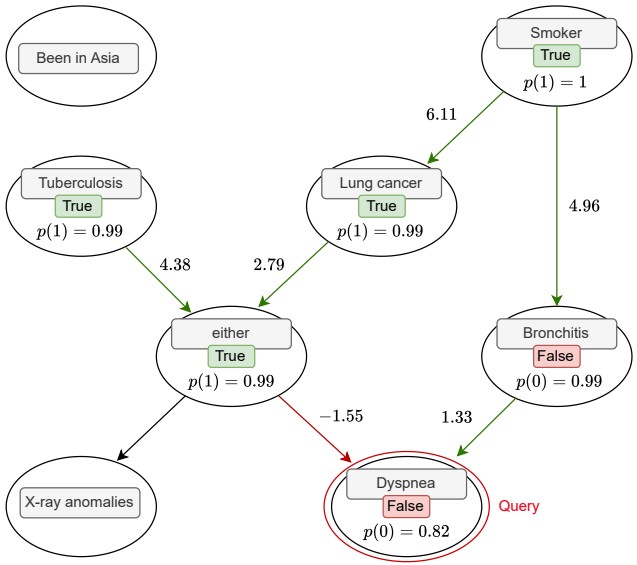

Figure 9: Visualization example of information propagation (Asia dataset). The parent weight (predicted by the hypernetwork) is visualized next to each edge.

# E Dataset details

## E.1 cMNIST

The MNIST dataset (LeCun et al., 2010) is a large collection of freely available grayscale images of handwritten digits. It consists of 60000 training images and 10000 test images, both drown from the same distribution. Each image is labeled with the digit it represents. For our experiments, we download the original train and test dataset using the *torchvision* library (Marcel & Rodriguez, 2010) and reserve 10% of the training set for validation. Additionally, we colorize each image based on its digit. Specifically, in the in-distribution setting (experiment in Sec.5.1), color is randomly assigned to each image in the training, validation, and test sets with equal probability over red, green, and blue. In the out-of-distribution setting (experiment in Sec.5.3), images in the training and validation sets with odd digits are colored green, while the remaining images are colored red or blue with

equal probability. In the test set, images with even digits are colored green, and the remaining ones are colored red or blue with equal probability. For both the versions of this dataset, the following concepts are considered: *Number*, *Color*, and *Parity* (task). Finally, the images are preprocessed using a pre-trained ResNet-18 model with default weights from the torchvision library.

## E.2  Bayesian networks

For our experiments, we use synthetic datasets sampled from discrete Bayesian networks available in the `bnlearn` repository (`https://www.bnlearn.com/bnrepository/`). A *Bayesian network* is a probabilistic graphical model consisting of a DAG, where nodes correspond to random variables, and each node is associated with a conditional probability distribution (CPD). This CPD defines the probability of the node's value, given the values of its parent nodes in the network (Sharma et al., 2020). From the `bnlearn` repository, we select Bayesian networks with different dimensions and domains: **Asia** (Lauritzen & Spiegelhalter, 1988), a small network focused on lung disease with 8 nodes and 8 edges; **Sachs** (Sachs et al., 2005), a widely-used network modeling the relationships between protein and phospholipid expression levels in human cells with 11 nodes and 17 edges; **Insurance** (Binder et al., 1997), a network for evaluating car insurance risks with 27 nodes and 52 edges; **Alarm** (Beinlich et al., 1989), a network designed to provide an alarm message system for patient monitoring with 37 nodes and 46 edges; **Hailfinder** (Abramson et al., 1996), a network designed to forecast severe summer hail in northeastern Colorado with 56 nodes and 66 edges. For each network, we generate 10000 samples and create training, validation, and test datasets using a $70\% - 10\% - 20\%$ split.

While node values can be used as concept annotations ($\mathbf{v}$), input features ($\mathbf{x}$) are absent. To generate them and make the datasets applicable to concept-based architectures, we flatten the concept values and process them with a simple autoencoder (MSE loss) comprising 2 encoder layers and 2 decoder layers (the latent dimension is adjusted based on the number of nodes: *Asia*-32, *Sachs*-32, *Insurance*-32, *Alarm*-64, *Hailfinder*-128). Embeddings are further transformed so that each sample is a mixture composed of $50\%$ original data and $50\%$ noise, with the noise drawn from a standard normal distribution. Finally, the output is standardized. The goal of these transformations is to make the inputs non-trivial representations of the concepts (the network nodes), forcing architectures to learn how to identify and retrieve the underlying concepts from the preprocessed data.

Modified versions of the *Asia* and *Alarm* datasets are considered, denoted as Asia* and Alarm*, in which only a subset of the original concepts is retained (experiment in Sec.5.1). Specifically, for *Asia**, we keep only the concepts "Smoke" and "Dyspnea". For *Alarm**, we retain only the concepts: "BP", "CO", "CATECHOL", "HR", "LVFAILURE", "STROKEVOLUME", "HYPOVOLEMIA".

## E.3  CelebA

CelebA (Liu et al., 2015) is a large-scale face attributes dataset with more than 200.000 celebrity images divided into training, validation and test set with 40 binary attribute annotations. For our experiments, we first downloaded all the splits from the project website `https://mmlab.ie.cuhk.edu.hk/projects/CelebA.html`.
We then select a subset of attributes that we consider relevant for our analysis and apply this selection to all the splits. These attributes are: *Attractive*, *Big Lips*, *Heavy Makeup*, *High Cheekbones*, *Male*, *Mouth Slightly Open*, *Oval Face*, *Smiling*, *Wavy Hair*, *Wearing Lipstick* (experiment in Sec.5.1). For our fairness analysis (experiment in Sec.5.4), we selected the attributes: *Attractive*, *Heavy Makeup*, *High Cheekbones*, *Male*, *Mouth Slightly Open*, *Oval Face*, *Pointy Nose*, *Smiling*, *Wavy Hair*, *Wearing Lipstick* and *Young*. We then introduced two additional new attributes *Qualified* and *Should be Hired*. In this analysis, we consider a hypothetical scenario in which a person with a pointy nose is required for a specific role, e.g., in a movie. However, the

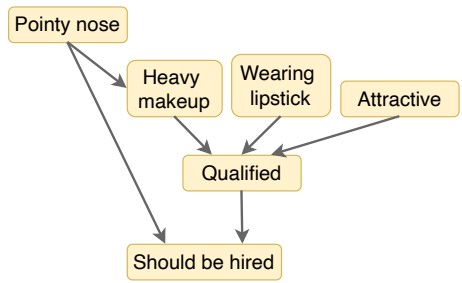

Figure 10: Subset of introduced causal relationships among original and newly created concepts in CelebA, used for our fairness analysis.

hiring manager has a strong bias toward models
who are considered attractive or heavily made up. Our aim is to intervene and mitigate such biases. The *Qualified* attribute is defined as a binary variable indicating whether a person meets the qualifications for the job based on the hiring manager's biased criteria. It is therefore constructed using the logical expression: (*Heavy Makeup* and *Wearing Lipstick*) or *Attractive*. The *Should be Hired* attribute, on the other hand, indicates whether a person should be hired for the job, considering both the hiring manager's preferences and the role's requirements (having a pointy nose). Therefore, it is defined as the logical "and" between *Qualified* and *Pointy Nose*. Additionally, we applied standard preprocessing to both versions of the dataset, including downsampling and normalization of the images, followed by feature extraction using a pre-trained ResNet-18 model.

### E.4 Siim-pneumothorax

This dataset is derived from the publicly available chest radiograph dataset provided by the National Institutes of Health (NIH) and it contains chest x-ray images with binary annotations indicating the presence or the absence of Pneumothorax. For our experiments, we use in particular the training annotations available on Kaggle (`https://www.kaggle.com/competitions/siim-acr-pneumothorax-segmentation`) and the corresponding training images from `https://www.kaggle.com/datasets/abhishek/siim-png-images`. The dataset is then split into training, validation, and test sets using a traditional $70\% - 10\% - 20\%$ partition. As it does not include concept annotations, we followed a procedure inspired by the methodology in Oikarinen et al. (2023) to generate concepts and their corresponding annotations. More details can be found in App. B.

### E.5 CUB$_C$

This dataset is derived from the publicly available Caltech-UCSD Birds-200-2011 (CUB) (He & Peng, 2019) dataset, which is widely used in the CBM community (Koh et al., 2020; Zarlenga et al., 2022). It contains 11,788 images across 200 bird categories, with 5,994 images for training and 5,794 images for testing. Each image is annotated in detail, including 312 binary attributes. For our experiments, we downloaded the dataset from `https://data.caltech.edu/records/65de6-vp158` and further split the training set such that $10\%$ is used for validation. We then selected the 112 most frequently activated binary attributes to serve as our concepts of interest, as considered in Zarlenga et al. (2022).

To explore deeper causal relationships between concepts, we introduce four new ones: *camouflage*, *flight adaptation*, and *hunting ability*, which are derived from existing attributes using logical rules, as well as the multi-valued concept *survival*, whose activation is in turn derived from these three binary concepts. The rules used for these derivations are detailed in the table below (Table 4). We use *survival* as our downstream task. The images are instead further downsampled, normalized, and processed using a ResNet-50 architecture, following a procedure similar to that used for the CelebA dataset.

## F  Experimental details

**Python code and instructions for reproducing results across all datasets and methods are available within the code provided alongside the submission as supplementary material.** We detail below the configurations and hyperparameters used for models' instantiation and training. For a fair comparison across concept-based models, we standardize the concept encoder to a single-hidden-layer MLP. All models are trained using the Adam optimizer (Kingma & Ba, 2015) for a maximum of 500 epochs, with early stopping based on a 30-epoch patience. We employ *LeakyReLU* as the activation function throughout.

The batch size is set to 512 for most datasets, with the exception of *Siim-pneumothorax* and SCBM, where it is reduced to 128 due to memory constraints. Following Koh et al. (2020), we regularize the task loss to encourage concept learning, using a weighted sum of task and concept losses:

$$L = (1 - \alpha) \cdot L_{\text{task}} + \alpha \cdot L_{\text{concepts}}, \quad \text{with } \alpha = 0.8$$

where $L_{\text{task}}$ is a cross-entropy over the task whereas $L_{\text{concepts}}$ is the summation of cross-entropy losses over the concepts.

Additionally, we apply random training-time interventions as proposed by Zarlenga et al. (2022), with an intervention probability of 0.25. Regarding SCBM, we used the authors' implementation

Table 4: Newly introduced CUB concepts and the rules used to determine their values.

| New Concept | Logical rules |
|---|---|
| camouflage | has_tail_pattern_spotted ∨
has_tail_pattern_striped ∨
has_tail_pattern_multi-colored ∨
has_back_pattern_spotted ∨
has_back_pattern_striped ∨
has_back_pattern_multi-colored |
| flight_adaptation | has_tail_shape_rounded_tail ∨
has_wing_shape_rounded-wings ∨
has_size_medium |
| hunting_ability | has_bill_shape_curved ∨
has_bill_shape_needle ∨
has_bill_shape_spatulate ∨
has_bill_shape_all-purpose ∨
has_bill_shape_longer_than_head ∨
has_bill_shape_shorter_than_head |
| survival | max(camouflage + flight_adaptation + hunting_ability, 2) |

at `https://github.com/mvandenhi/SCBM`. More precisely, we implemented the *global* variation using the configuration proposed by the authors.

Key hyperparameters, including learning rate, MLP hidden size, and dropout rate, are selected via grid search. A complete list of hyperparameters for $C^2BM$ and all baseline models can be found in the provided YAML configuration files within the code.

All experiments are conducted on NVIDIA GeForce RTX 3080 and NVIDIA RTX A5000 GPUs.

## F.1 Metric: custom Structural Hamming Distance

To assess the quality of the causal graphs generated by our pipeline and baseline discovery models, we employ two metrics: a variant of the structural Hamming distance (SHD)[9] that operates on CPDAGs, and the number of incorrect edges identified. The number of incorrect edges serves as a standard measure of the quality of the graph, while the SHD allows us to customize weights for different types of errors. Tab. 5 provides a schematic of our SHD scores:

| Ground Truth | Predicted | SHD Penalty |
|:---:|:---:|:---:|
| / | $i \to j$ | 1 |
| / | $i - j$ | 1/2 |
| $i \to j$ | $i \leftarrow j$ | 1/3 |
| $i \to j$ | / | 1/4 |
| $i - j$ | / | 1/4 |
| $i - j$ | $i \to j$ | 1/4 |
| $i \to j$ | $i - j$ | 1/5 |

Table 5: SHD penalties for various discrepancies between ground truth and predicted edges. '$i \to j$': oriented edge, '$i - j$': unoriented edge, '/': no edge

The rationale behind the scores is that the insertion of new edges is 'riskier' than the removal of existing ones. The introduction of non-existing edges may induce spurious correlations that strongly affect the reliability of the model and related metrics, such as counterfactual fairness or accuracy in ood tasks. On the contrary, removing an existing edge is a more conservative operation: while it still

---

[9]Although we refer to this as a "distance," it is technically an asymmetric scoring function.

affects the model accuracy, is not strongly impacting reliability. For the same reason, introducing an incorrect edge orientation is more penalized than removing an orientation.

## G   Additional experiments

### G.1   Label accuracy

We report here the average label accuracy (task + concepts) for each of the datasets analyzed in Tab. 1. In this setting, since we are evaluating the accuracy across both the task and the concepts, we use a different opaque neural baseline, $OpaqNN_M$, which jointly predicts all concepts and the task for each dataset. As shown in the Tab. 6, $C^2BM$ achieves comparable results to non-causal models in terms of concept prediction. Notably, the performance differences observed in Section 5.1 for task accuracy are attenuated here due to averaging over both tasks and concepts.

Table 6: Label accuracy (%). Task concepts are as follows: dysp (Asia), Akt (Sachs), BP (Alarm), PropCost (Insurance), R5Fcst (Hailfinder), parity (cMNIST), mouth slightly open (CelebA), Survival ($CUB_C$), pneumothorax (Pneumoth.). Uncertainties represent 2 sample mean $\sigma$ across 5 runs.

| Model | Semantic Transp. | Causal Rel. | Asia | Sachs | Insurance | Alarm | Hailfinder | cMNIST | CelebA | $CUB_C$ | Pneumoth. |
|---|---|---|---|---|---|---|---|---|---|---|---|
| $OpaqNN_M$ | ✗ | ✗ | $87.1_{\pm1.1}$ | $72.4_{\pm1.0}$ | $78.76_{\pm0.73}$ | $90.13_{\pm0.30}$ | $66.29_{\pm0.60}$ | $91.85_{\pm.44}$ | $80.30_{\pm0.04}$ | $77.66_{\pm0.49}$ | $74.99_{\pm0.34}$ |
| $CBM_{+lin}$ | ✔ | ✗ | $87.0_{\pm1.0}$ | $71.9_{\pm1.2}$ | $78.65_{\pm0.68}$ | $89.88_{\pm0.30}$ | $69.99_{\pm0.67}$ | $91.85_{\pm.44}$ | $79.98_{\pm0.14}$ | $77.51_{\pm0.43}$ | $74.03_{\pm0.44}$ |
| $CBM_{+mlp}$ | ✔ | ✗ | $87.1_{\pm0.9}$ | $72.2_{\pm1.2}$ | $78.64_{\pm0.70}$ | $89.80_{\pm0.36}$ | $69.78_{\pm0.79}$ | $91.46_{\pm.40}$ | $80.06_{\pm0.03}$ | $77.75_{\pm0.54}$ | $73.72_{\pm0.65}$ |
| CEM | ✔ | ✗ | $87.1_{\pm0.9}$ | $72.2_{\pm1.1}$ | $78.60_{\pm0.77}$ | $89.77_{\pm0.28}$ | $70.14_{\pm0.76}$ | $91.42_{\pm.37}$ | $80.39_{\pm0.02}$ | $77.16_{\pm0.33}$ | $74.27_{\pm0.77}$ |
| SCBM | ✔ | ✗ | $87.0_{\pm0.9}$ | $72.2_{\pm1.0}$ | $78.62_{\pm0.75}$ | $90.08_{\pm0.32}$ | $69.54_{\pm0.65}$ | $91.84_{\pm.37}$ | $79.93_{\pm0.03}$ | $78.03_{\pm0.23}$ | $74.31_{\pm0.29}$ |
| $C^2BM$ | ✔ | ✔ | $87.2_{\pm1.0}$ | $72.0_{\pm0.9}$ | $78.37_{\pm0.72}$ | $89.83_{\pm0.35}$ | $71.85_{\pm0.80}$ | $92.19_{\pm.05}$ | $80.44_{\pm0.10}$ | $77.22_{\pm0.36}$ | $74.73_{\pm0.31}$ |

### G.2   Task accuracy using true graph

To further assess the quality of the generated graph, we compare downstream task performance using the inferred graph with performance using the true graph, available in Bayesian Network synthetic datasets.

Table 7: Task accuracy (%) using the *true* and *predicted* graph. Task concepts are as follows: dysp (Asia), Akt (Sachs), PropCost (Insurance), BP (Alarm), R5Fcst (Hailfinder). Uncertainties represent 2 sample mean $\sigma$ across 5 runs.

| Model | | Asia | Sachs | Insurance | Alarm | Hailfinder |
|---|---|---|---|---|---|---|
| $C^2BM$ | True graph | $71.0_{\pm2.3}$ | $65.0_{\pm1.3}$ | $66.4_{\pm1.8}$ | $62.1_{\pm1.5}$ | $73.0_{\pm1.6}$ |
| | Predicted graph | $71.4_{\pm1.7}$ | $65.3_{\pm1.1}$ | $66.4_{\pm1.5}$ | $62.5_{\pm1.4}$ | $74.1_{\pm1.8}$ |

Results in Tab. 7 indicate that the two settings yield comparable performance, demonstrating both the quality of the learned graph and the robustness of the proposed pipeline. This robustness can be attributed to the employed exogenous latent embeddings, which mitigate the impact of concept incompleteness. Consequently, even when the inferred graph is not perfectly aligned with the true causal structure, $C^2BM$ maintains strong task performance. These findings highlight the model's resilience and its ability to generalize effectively in real-world scenarios where true causal structures are often unavailable.

### G.3   Ablation study on Causal Discovery

In this section, we evaluate the sensitivity of the causal discovery component in our pipeline to the choice of method for constructing a causal graph. To do so, we compare our method of choice for causal discovery, i.e., *Greedy Equivalence Search* (GES) (Chickering, 2002), with different widely-used causal discovery algorithms that recover a CPDAG. Each of these methods is evaluated with and without refinement via the retrieval-augmented generation (RAG)-enhanced language model

(LLM) employed in our study. Furthermore, we compare all these methods against the use of the LLM alone, as well as our retrieval-augmented LLM (LLM + RAG) applied directly to discover the whole graph. Specifically, the LLM is prompted with the causal prompt shown in App. C.3, with additional retrieved context appended to the query in the LLM+RAG setting.

Specifically, we evaluate the following methods from the causal discovery literature:

- *GES* (score-based): The algorithm selected in our study. A detailed description is provided in App. C;
- *PC Algorithm* (Spirtes et al., 2000) (independence-based): A classical independence-based method that first estimates the undirected structure of the causal relations through a sequence of conditional independence tests, then orients edges using a set of predefined orientation rules (Colombo et al., 2014). In our implementation, we use the PC algorithm provided by the `causal-learn` library (Zheng et al., 2024), with a chi-squared independence test and a significance level of 0.05;
- *Fast GES* (Ramsey et al., 2017) (FGES) (score-based): A computationally efficient variant of GES that improves performance by storing intermediate evaluations and parallelizing expensive operations, enabling its application to large datasets (Andrews et al., 2019; Ramsey et al., 2017). In our implementation, we use the FGES algorithm provided by the `py-tetrad` library (Ramsey & Andrews, 2023), employing the *Bayesian Dirichlet Equivalent Uniform* (BDeu) score for structure evaluation (Heckerman et al., 1995).

The results are evaluated on all the `bnlearn` datasets, for which the true causal graphs are known. Results are presented in Tab. 8.

Table 8: Structural Hamming distance and number of mistaken edges between the true and learned causal graphs for each tested method. The maximum number of errors (Max errors) and the average number of errors make by a random classifier (Random Classifier) are also provided for reference.

| Metric | CD Method | Asia | Sachs | Insurance | Alarm | Hailfinder |
|---|---|---|---|---|---|---|
| Hamming | LLM | 13 | 11.08 | 336 | 505.92 | 92.75 |
| | LLM + RAG | 12 | 6.83 | 201.91 | Time limit | Time limit |
| | PC | 2.41 | 2.93 | 6.65 | 5 | 14.53 |
| | FGES | 0.65 | 3.4 | 6.78 | 5.91 | 21.4 |
| | GES | 0.65 | 3.4 | 6.43 | 5.41 | **11** |
| | PC + LLM (+ RAG) | 2.41 | 2.91 | 6 | **3.92** | 15.42 |
| | FGES + LLM (+ RAG) | **0.25** | 2.03 | **5.33** | 6 | 20.5 |
| | GES + LLM (+ RAG) | **0.25** | **1.83** | 6.33 | 4.95 | **11** |
| Number of mistaken edges | Max errors | 28 | 55 | 351 | 666 | 1540 |
| | Random Classifier | 9.33 | 18.33 | 117 | 222 | 513.33 |
| | LLM | 13 | 20 | 351 | 543 | 107 |
| | LLM + RAG | 12 | 12 | 226 | Time limit | Time limit |
| | PC | 6 | 10 | 26 | 13 | 45 |
| | FGES | 3 | 17 | 23 | 14 | 39 |
| | GES | 3 | 17 | 19 | 13 | **22** |
| | PC + LLM (+ RAG) | 6 | 9 | 22 | **9** | 43 |
| | FGES + LLM (+ RAG) | **1** | 8 | **15** | 11 | 36 |
| | GES + LLM (+ RAG) | **1** | **7** | 18 | 10 | **22** |

As shown in the results, GES refined with the RAG-augmented LLM is the overall best-performing method in the majority of cases, both in terms of structural Hamming distance and number of mistaken edges. In instances where it is not the best, it still consistently ranks among the top methods. Notably, both the standard causal discovery method (GES) and the use of the RAG-augmented LLM contribute positively to performance. Due to the substantial computational time required by the LLM+RAG-based causal discovery approach, we excluded experiments on datasets for which execution exceeded 24 hours.

## G.4 Ablation study on LLM type

The LLM or the design of the LLM prompt could condition the causal graph refinement. To assess this, we present an ablation study in which we fix the incomplete causal graph generated by the causal

discovery algorithm (GES), and later evaluate different LLMs, with different prompts, for the graph refinement step. Specifically, we evaluate the impact of four different prompting strategies with three LLMs (GPT-4o, 200M parameters; and GPT-4o-mini, 8M parameters). The considered strategies are as follows:

- *Minimal* prompting: simply asks the LLM to identify causal relations without additional guidance;
- *Instruction* prompting: provides a more detailed explanation of what constitutes a causal relation;
- *Few-shot* prompting: proposed in Ye et al. (2024), combines instruction prompting with a few examples;
- *Chain-of-Thought* (CoT): proposed in Wei et al. (2022), encourages the model to generate intermediate logical steps before generating the answer. This is the approach we used in the original manuscript.

Tab. 9 compares the number of mistaken edges and the custom Hamming distance between the true and the learned DAG. Causal reliability of standard, flat, CBMs is also reported for reference. The results show that, using GPT-5, causal reliability improves on average w.r.t. GPT-4o, and GPT-4o-mini, especially on the Sachs dataset. Despite moderate variation in the Sachs dataset, the performance is generally very robust to the prompt strategy (made an exception for naive minimal strategy), and causal reliability is largely superior to standard CBMs in all cases.

Table 9: Structural Hamming distance and number of Mistaken Edges between true and learned DAG across datasets, using different LLMs and prompting strategies. Uncertainty represents 2 sample mean $\sigma$ across 3 runs.

| Metric | LLM | Prompting Strategy | Asia | Sachs | Insurance | Alarm | Hailfinder |
|---|---|---|---|---|---|---|---|
| Hamming | GPT-4o-mini | Minimal | 0.25±.00 | 2.67±.00 | 8.04±.52 | 3.63±.94 | 10.83±.51 |
| | | Instruction | 0.25±.00 | 2.75±.06 | 8.04±.52 | 4.3±1.2 | 10.83±.51 |
| | | Few-shot | 0.25±.00 | 2.79±.05 | 8.04±.52 | 4.3±1.2 | 10.83±.51 |
| | | CoT | 0.25±.00 | 2.21±.08 | 7.67±.48 | 4.1±1.1 | 10.83±.51 |
| | GPT-4o | Minimal | 0.25±.00 | 2.83±.00 | 8.04±.52 | 3.75±.98 | 10.83±.51 |
| | | Instruction | 0.25±.00 | 2.63±.16 | 8.04±.52 | 4.7±1.3 | 10.83±.51 |
| | | Few-shot | 0.25±.00 | 2.46±.05 | 8.04±.52 | 4.8±1.3 | 10.83±.51 |
| | | CoT | 0.25±.00 | 2.25±.06 | 7.54±.52 | 4.7±1.3 | 10.83±.51 |
| | GPT-5 | Minimal | 0.25±.00 | 1.29±.05 | 7.92±.48 | 4.2±1.1 | 10.83±.51 |
| | | Instruction | 0.25±.00 | 1.21±.02 | 7.29±.43 | 4.4±1.2 | 10.83±.51 |
| | | Few-shot | 0.25±.00 | 1.04±.05 | 7.29±.43 | 4.0±1.1 | 10.83±.51 |
| | | CoT | 0.25±.00 | 0.92±.09 | 7.17±.39 | 4.6±1.3 | 10.83±.51 |
| Mistaken Edges | GPT-4o-mini | Minimal | 1.00±.00 | 9.00±.00 | 23.0±1.4 | 9.0±1.8 | 21.50±.90 |
| | | Instruction | 1.00±.00 | 9.50±.18 | 23.0±1.4 | 10.0±2.2 | 21.50±.90 |
| | | Few-shot | 1.00±.00 | 9.50±.18 | 23.0±1.4 | 10.0±2.2 | 21.50±.90 |
| | | CoT | 1.00±.00 | 8.00±.36 | 21.5±1.3 | 9.0±1.8 | 21.50±.90 |
| | GPT-4o | Minimal | 1.00±.00 | 11.00±.00 | 23.0±1.4 | 9.5±2.0 | 21.50±.90 |
| | | Instruction | 1.00±.00 | 10.00±.72 | 23.0±1.4 | 10.0±2.2 | 21.50±.90 |
| | | Few-shot | 1.00±.00 | 9.50±.18 | 23.0±1.4 | 11.0±2.3 | 21.50±.90 |
| | | CoT | 1.00±.00 | 8.50±.18 | 21.0±1.4 | 10.0±2.2 | 21.50±.90 |
| | GPT-5 | Minimal | 1.00±.00 | 4.50±.18 | 22.5±1.3 | 9.5±2.0 | 21.50±.90 |
| | | Instruction | 1.00±.00 | 4.00±.00 | 20.0±1.1 | 9.0±1.8 | 21.50±.90 |
| | | Few-shot | 1.00±.00 | 3.50±.18 | 20.0±1.1 | 9.0±1.8 | 21.50±.90 |
| | | CoT | 1.00±.00 | 3.00±.36 | 19.5±.9 | 9.5±2.0 | 21.50±.90 |

## G.5 Ablation study on RAG

Tab. 10 illustrates the influence of the context supplied by RAG in correcting the causal graph generated by the causal discovery algorithm. In this analysis, we used `GPT-4o` for both the experiments with and without RAG, aiming to evaluate the effect of context on answering causal queries.

Although the context provided by RAG appears to have no significant impact on the final causal graph for the Alarm dataset, it proves essential for correctly handling the undirected edges in the causal graph for the Sachs dataset. We hypothesize that this is due to the absence of protein related documents used in training the LLM, which leaves it with insufficient prior knowledge to address

Table 10: Structural Hamming distance and number of mistaken edges between true and learned DAG when using either RAG to provide context to the LLM or just the LLM.

| Metric | Context | Sachs | Alarm |
|--------|---------|-------|-------|
| Hamming | No context | 3.1 | **5.0** |
| | RAG context | **1.8** | **5.0** |
| Mistaken edges ratio | No context | 12 | **10** |
| | RAG context | **7** | **10** |

specific questions on the topic. RAG helps mitigate this limitation by supplying the LLM with the relevant information, thereby compensating for the lack of prior knowledge. In conclusion, while an LLM with the ability to fully comprehend complex queries is crucial for causal discovery, the additional context provided by RAG is vital for overcoming the LLM's prior knowledge gaps.

### G.6 Sample complexity of graph discovery

We study the effect of dataset size and graph size on the quality of $C^2BM$'s graph construction pipeline. We explored this with a sensitivity study, running the causal graph pipeline (causal discovery with GES + LLM refinement with CoT prompt) across all datasets with an available ground-truth graph and comparing the number of mistaken edges between the true and the learned DAG. For each dataset, we varied the number of data points $N$ from 100 to 10000. Results are presented in Tab. 11.

Table 11: Number of mistaken edges between the true DAG and the learned graph using the $C^2BM$'s graph construction pipeline, evaluated across datasets and data sizes. For reference, the reliability of standard flat CBMs is also reported.

| Dataset | Flat CBMs | $C^2BM$'s causal graph | | | | |
|---------|-----------|-----|-----|------|------|---------------|
| | | Data size ($N$) $\rightarrow$ 100 | 500 | 1000 | 5000 | 10000 (Paper) |
| Asia | 11 | 6 | 3 | 1 | 1 | 1 |
| Sachs | 23 | 15 | 12 | 11 | 7 | 7 |
| Insurance | 74 | 47 | 35 | 24 | 22 | 18 |
| Alarm | 78 | 31 | 16 | 9 | 9 | 9 |
| Hailfinder | 117 | 56 | 47 | 44 | 22 | 22 |

A few considerations emerge:

- $C^2BM$'s causal graph is consistently more causally reliable than the flat structure implicitly assumed by standard CBMs, regardless of the dataset size.

- As expected, increasing the number of data points leads to better alignment between the estimated and true causal graphs. Causal reliability tends to remain stable at larger data sizes.

- We observed that the data size threshold for reliable performance does not strictly depend on graph size. We speculate this is due to the varying impact of LLM-based refinement across datasets. In some cases, an effective background knowledge can compensate for limited data.

### G.7 Sensitivity to graph corruption

In this section, we empirically assess the robustness of $C^2BM$ to graph misspecification and corruption. Although $C^2BM$ is theoretically a universal approximator for the final prediction task, independent of the specific causal graph (see Appendix D.1), we complement this result with an empirical validation.

- **Adversarial Graph Corruptions.** We first evaluate robustness by altering a percentage $p$ of graph edges, chosen randomly, with one of the following operations: *edge flipping*, *addition*,

or *removal*. The resulting performance across datasets and corruption levels is reported in Tab. 12.

- **Progressive Flattening into Standard CBMs.** As a second evaluation, we progressively transform the graph into the flat structure assumed by standard CBMs, by connecting a percentage $p$ of nodes directly to the prediction task while removing their outgoing edges. Results are shown in Tab. 13.

Table 12: Task accuracy (%) under edge-level adversarial corruptions. A percentage $p$ of edges is altered through flipping, addition, or removal. Task concepts are as follows: dysp (Asia), Akt (Sachs), PropCost (Insurance), BP (Alarm), R5Fcst (Hailfinder). Uncertainties represent 2 sample mean $\sigma$ across 3 runs.

| Dataset / $p$ | 0.05 | 0.1 | 0.2 | 0.4 | 0.6 | 0.8 | 1.0 |
|---|---|---|---|---|---|---|---|
| Asia | $71.8_{\pm 0.6}$ | $71.6_{\pm 0.5}$ | $71.4_{\pm 0.6}$ | $71.7_{\pm 1.3}$ | $70.2_{\pm 1.9}$ | $71.4_{\pm 1.2}$ | $70.2_{\pm 0.9}$ |
| Sachs | $65.1_{\pm 1.9}$ | $65.6_{\pm 2.0}$ | $64.9_{\pm 2.0}$ | $64.6_{\pm 1.2}$ | $65.5_{\pm 1.5}$ | $65.2_{\pm 1.5}$ | $65.0_{\pm 2.0}$ |
| Insurance | $67.2_{\pm 1.6}$ | $67.0_{\pm 3.1}$ | $67.5_{\pm 2.3}$ | $66.1_{\pm 1.1}$ | $67.0_{\pm 2.6}$ | $66.8_{\pm 2.6}$ | $67.2_{\pm 3.1}$ |
| Alarm | $61.9_{\pm 2.6}$ | $61.7_{\pm 2.1}$ | $61.5_{\pm 2.5}$ | $61.9_{\pm 2.7}$ | $60.6_{\pm 1.9}$ | $60.9_{\pm 1.6}$ | $61.4_{\pm 1.7}$ |
| Hailfinder | $74.0_{\pm 1.5}$ | $73.1_{\pm 3.0}$ | $72.3_{\pm 2.2}$ | $72.3_{\pm 3.2}$ | $73.0_{\pm 2.1}$ | $71.8_{\pm 1.6}$ | $72.4_{\pm 1.4}$ |

Table 13: Task accuracy (%) under progressive graph flattening into standard CBMs. A percentage $p$ of nodes is directly connected to the task output. Task concepts are as follows: dysp (Asia), Akt (Sachs), PropCost (Insurance), BP (Alarm), R5Fcst (Hailfinder). Uncertainties represent 2 sample mean $\sigma$ across 3 runs.

| Dataset / $p$ | 0.05 | 0.1 | 0.2 | 0.4 | 0.6 | 0.8 | 1.0 |
|---|---|---|---|---|---|---|---|
| Asia | $72.0_{\pm 1.0}$ | $72.0_{\pm 1.0}$ | $71.5_{\pm 0.3}$ | $71.7_{\pm 0.8}$ | $71.7_{\pm 0.5}$ | $71.2_{\pm 1.6}$ | $71.1_{\pm 1.1}$ |
| Sachs | $65.5_{\pm 2.2}$ | $64.8_{\pm 3.0}$ | $65.8_{\pm 1.6}$ | $64.7_{\pm 1.5}$ | $65.2_{\pm 1.2}$ | $65.1_{\pm 2.7}$ | $64.9_{\pm 2.2}$ |
| Insurance | $67.4_{\pm 2.6}$ | $66.2_{\pm 1.7}$ | $66.3_{\pm 2.5}$ | $65.6_{\pm 3.3}$ | $65.3_{\pm 2.4}$ | $66.9_{\pm 1.8}$ | $64.7_{\pm 3.1}$ |
| Alarm | $62.6_{\pm 2.9}$ | $61.6_{\pm 2.6}$ | $61.6_{\pm 2.0}$ | $60.5_{\pm 2.6}$ | $61.2_{\pm 1.8}$ | $60.6_{\pm 3.2}$ | $61.3_{\pm 1.9}$ |
| Hailfinder | $73.1_{\pm 1.1}$ | $73.2_{\pm 1.9}$ | $72.5_{\pm 0.9}$ | $72.7_{\pm 2.9}$ | $73.0_{\pm 2.6}$ | $73.0_{\pm 1.7}$ | $72.9_{\pm 1.9}$ |

Across both corruption strategies, we find that task accuracy **remains stable**, even when the causal graph is heavily perturbed. These empirical results support the theoretical claim that $C^2BM$ is robust to graph misspecification.

### G.8 Effect of single-concept interventions

Fig. 11 presents the relative improvement in task accuracy after intervening on individual concepts across all datasets. A key observation is that **$C^2$BM responds to interventions on the same key concepts as the baselines, despite the fundamental difference in how information propagates**. In CBM-based models and CEM, all concepts are directly connected to the task, enabling direct influence. In contrast, $C^2$BM enforces information flow through the causal graph, constraining the interactions. Yet, the task performance improvements remain consistent across models. These results highlight that $C^2$BM preserves the intervention effects observed in traditional concept-based models while providing a more structured and interpretable causal representation of the underlying relationships.

### G.9 Decomposing interventional accuracy

Fig. 4 in the main paper illustrates the improvement in cumulative relative interventional accuracy across all downstream, non-intervened concepts, including both intermediate concepts and the final task. To further analyze these effects, Fig. 12 decomposes this metric into two separate evaluations: one focusing solely on the task node and another considering only intermediate concepts. The concept

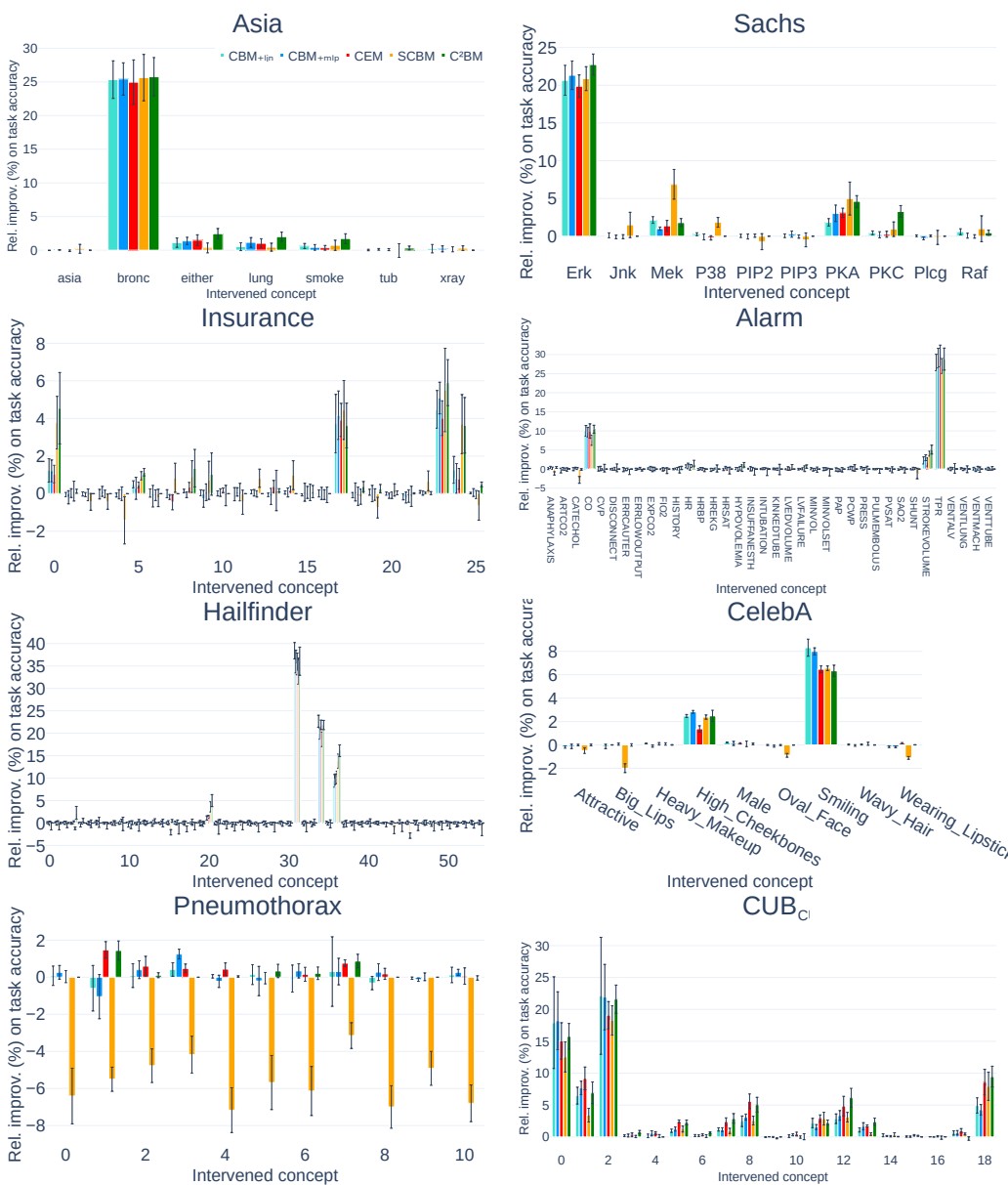

Figure 11: Relative improvement (%) in task accuracy when intervening on specific concepts.

accuracy plots highlight $C^2BM$'s unique ability to enhance performance of intermediate (downstream) concepts, a property not observed in competing models.

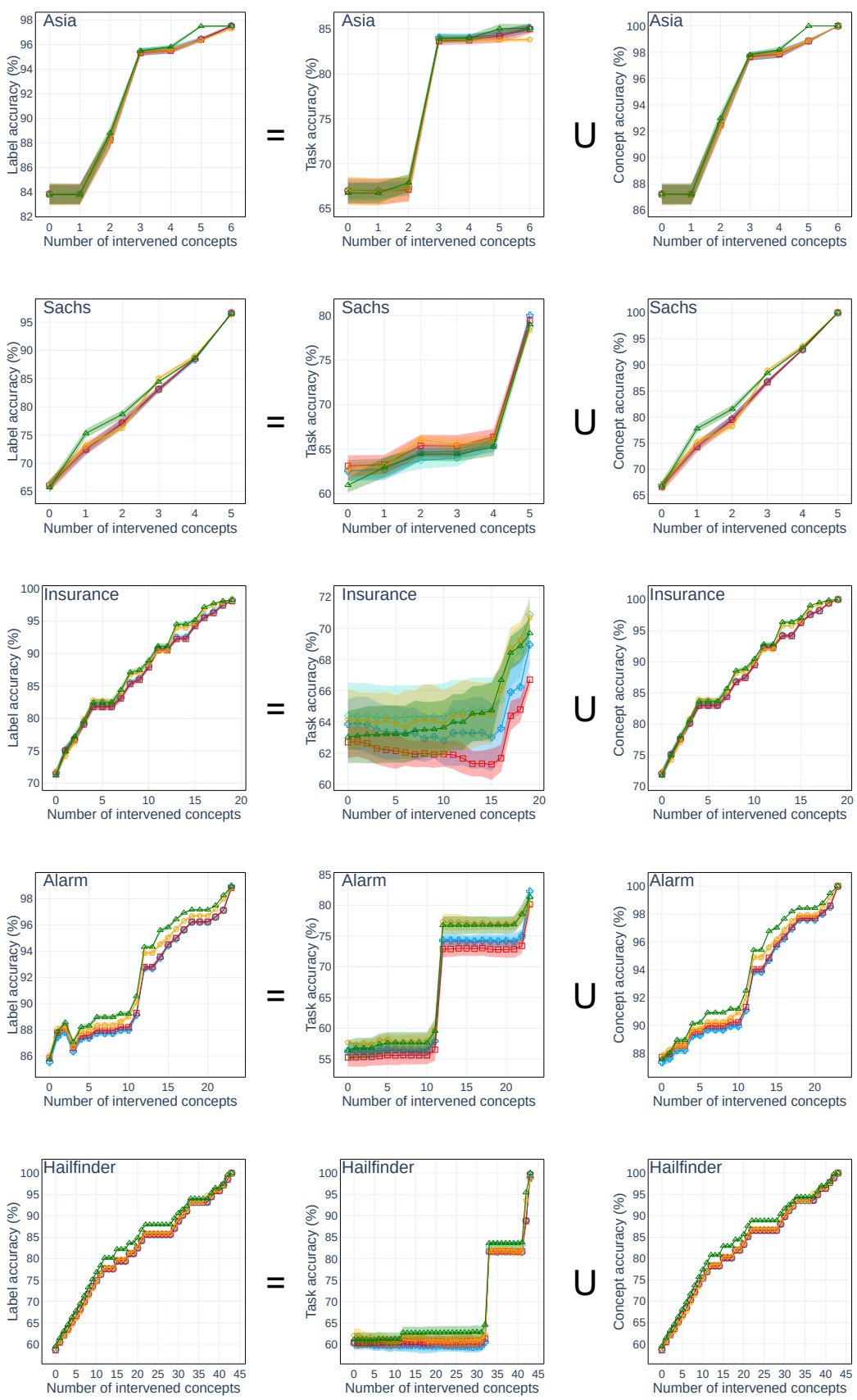

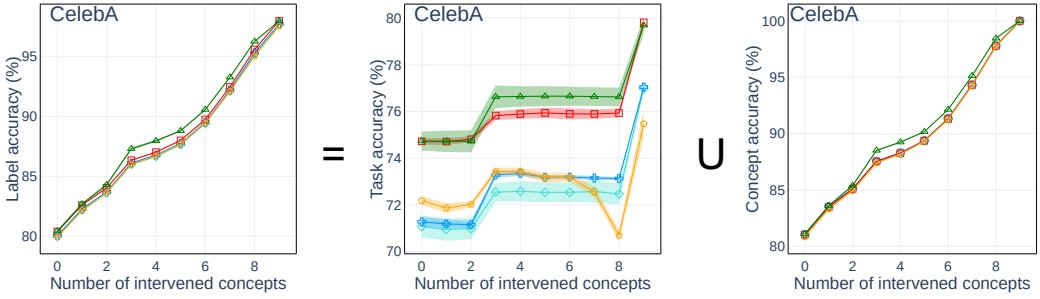

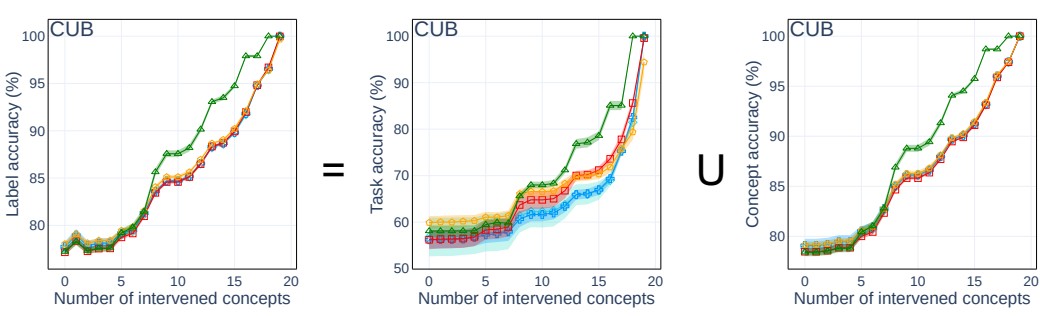

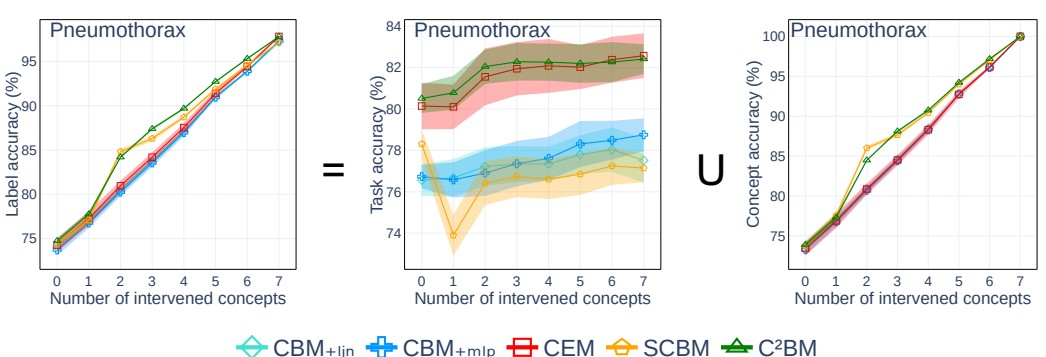

Figure 12: Task accuracy improvement (%) in predicting downstream variables after intervening on groups of concepts up to progressively deeper levels in the graph hierarchy. The metric is averaged across all downstream variables (Left), the task only (Middle), and all concepts (Right) Total label accuracy. Uncertainties represent 2 sample mean $\sigma$ across 5 runs.

