# OpenReview forum: "Causally Reliable Concept Bottleneck Models"
_NeurIPS.cc/2025/Conference — NeurIPS 2025 poster_

### Official Review · Reviewer_mU7E · 2025-06-25

**Clarity:** 2
**Significance:** 3
**Originality:** 3
**Rating:** 4
**Confidence:** 3

**Summary:**

This work attempts to bridge the gap between CBMs and Causal reliability with a new framework C2BM. It is based on performing LLM guided causal discovery on an identified concept space and combining this with structured causal modelling. The authors investigate the overall predictive performance as well as the. ability to perform concept interventions and debiasing.

**Questions:**

I would consider citing [1] in terms of concept interventions:

[1] Steinmann, David, Wolfgang Stammer, Felix Friedrich, and Kristian Kersting. "Learning to intervene on concept bottlenecks." arXiv preprint arXiv:2308.13453 (2023).

**Ethical Concerns:**

["NO or VERY MINOR ethics concerns only"]

**Final Justification:**

My clarification questions have been addressed. I have skimmed over the other reviews, but haven't seen any major issue which would change my view of my score.

**Limitations:**

Yes

**Quality:**

3

**Strengths And Weaknesses:**

I generally like the idea very much of combining concept XAI with causal reliability. Also the paper is very well written and it is easy to follow. Generally, I have very few weaknesses, just a few clarification issues which it would be good if the authors could clear up my misunderstandings. I am aware that the predictive performance experimental results don't look smashing, but I don't think this should be the focus of this work, i.e. the approach is interesting enough to overlook this.

Mainly, I am missing where the label prediction occurs in the method description, but also within figure 1. Maybe I am not an expert enough on SCMs, but how are the classifications of the evaluations done based on the setup, i.e. how is the most likely variable selected from the causal graph bottleneck. Standard CBMs utilize a final predictor, but I am missing the information on how this is handled here.

In this context I am still a little confused about what the goal is of C2BM and how this differs from simply doing causal discovery. It seems an image is first processed into concepts and then a causal graph is discovered based on this. Could the authors elaborate on this?

In this context I am also a little puzzled by the broader impact statement "C2BM has the potential to significantly narrow the hypothesis space in complex scientific domains where even a panel of human experts might struggle to identify or exclude plausible hypotheses worth testing. " again isn't this "simply" causal discovery? I.e. why do we need C2BM for this use case?

The debiasing experiment is a little unclear to me. How are the interventions being done here? I would assume if a vanilla CBM has color and shape concepts it should be very easy to debias the model, by reducing the reliance on the spurious concept. Could the authors clarify how exactly the results of figure 5 were created, particularly for the baseline models?

I'm a little surprised the effect of C2BMs in Fig. 4 is quite low (often only a few decimal point advantages). I would expect particularly here such models to shine. Could the authors discuss a little bit on this. Do these datasets simply don't require such interventions?

Tab. 1 is very squeezed. Do you think it might be an idea to move two columns to the appendix to improve the readability within the main text?

---

> ### Author Rebuttal · Authors · 2025-07-30
>
> We thank the reviewer for their comments and feedback. We particularly appreciate the positive remarks on the novelty, soundness, and clarity of our work. As for the raised questions, individual points are addressed below.
>
> ***Q1 - Label prediction***
>
> > I am missing where the label prediction occurs in the method description
>
> Starting from a data sample (e.g., raw image), we first extract a latent representation using a neural encoder. Then, an MLP is used to predict the probability of each root concept being in a specific state (e.g., true/false). Then, the state probabilities for all descendant concepts (including the task concept) are obtained by **iteratively applying Eq.2**. Specifically, Eq.2 shows that the probability of each concept state is determined by a linear combination of the state probabilities of its parents in the causal graph. Notice that, once the causal graph is determined and C$^2$BM is trained, one can choose any endogenous variable (concept) as the query (task).
> We refer to Fig. 7 in the Appendix for a more detailed visualization.
>
> We recognize the paper speaks in terms of 'variables' $V_i$ rather than state probabilities. This could be ambiguous. To improve clarity on this aspect, we added a footnote in l.193: "We refer to $V_i$ as a variable to allow for a more general formalization. In a classification setting, concepts assume categorical values; hence $V_i$ represents the probability of a concept state activation."
>
>
>
>
> ***Q2 - Difference with causal discovery***
>
> > the goal is of C$^2$BM and how this differs from simply doing causal discovery
>
> It is important to distinguish C$^2$BM from the components that need to be determined **prior** to its training and functioning. C$^2$BM is not a causal discovery algorithm. It is an inference ML model (see right side of Fig.1) trained on a given causal graph of concepts. For each data sample, the model learns to predict the activation of each variable in the graph based on its causal parents. In this setup, the causal graph acts as a structural prior, effectively a hyperparameter, which can be obtained from expert knowledge and/or from (potentially separate) data using causal discovery methods (on this, see Remark 4.1 and response Q2 to Reviewer wNgz).
>
> To give a concrete example: causal discovery might be used by a hospital to infer which factors are, in general, direct causes of a disease (as in our *Pneumothorax* dataset). C$^2$BM, once trained on this structure, can then make patient-specific predictions, e.g., estimating whether a particular patient is sick and which of the known causes are likely active in their case.
>
>
> ***Q3 - Clarification on broader impact***
>
> > I am also a little puzzled by the broader impact statement "C$^2$BM has the potential to significantly narrow the hypothesis space in complex scientific domains where even a panel of human experts might struggle to identify or exclude plausible hypotheses worth testing."
>
> We thank the reviewer for pointing this out; we recognize this sentence could be unclear. The sentence highlights a benefit commonly associated with causal discovery, one that C$^2$BM both **inherits and extends**. While causal discovery narrows the hypothesis space by uncovering candidate causal structures, C$^2$BM builds on top of this and enables predictive inference over those structures. That is, C$^2$BM allows for interpretable, concept-level predictions grounded in those relationships. In this way, C$^2$BM preserves the scientific value of causal discovery, while adding the practical ability to make predictions from data and unstructured knowledge, and generate explanations, critical in real-world decision-making contexts.
>
>
> ***Q4 - Clarification on debiasing experiment***
>
> > I would assume if a vanilla CBM has color and shape concepts it should be very easy to debias the model
>
> While, in principle, a standard CBM could support debiasing by manually setting the appropriate weights to zero, this requires knowing which relations are spurious in advance. In practical applications, this could require substantial domain expertise. In contrast, our pipeline incorporates a causal graph construction component that can **automatically detect and remove spurious correlations** from unstructured knowledge. C$^2$BM's inference flows through such graph that is already debiased, and hence does not require manual post-hoc debiasing.
>
> > How are the interventions being done here? ... Could the authors clarify how exactly the results of figure 5 were created
>
> The interventions in this experiment follow the standard practice in concept-based explainability [1]: we replace the predicted concept value with its ground-truth state and measure the consequent improvement in the task accuracy.
>
> Specifically, all models were trained on a biased training set (color $-$ parity) and evaluated on a test set where the spurious correlation is reversed. As expected, all models perform poorly ($\sim 0$%) when tested on reversed data. However, unlike the baselines, C$^2$BM is able to recover task accuracy when we intervene on the concept that is causally related to the task (number $\rightarrow$ parity). The other models fall short in doing so, since they still rely on spurious correlations.
>
>
> ***Q5 - Interventional accuracy***
>
> > I would expect particularly here such models to shine. Could the authors discuss a little bit on this.
>
> We speculate these are primarily due to the probabilistic nature of our datasets. In such settings, fixing the state of the parents is often **not sufficient** to uniquely determine the state of a child. **Exogenous factors also matter**. Therefore, achieving large improvements in intervention performance in these settings is inherently difficult. As shown in Fig.4, even baseline models exhibit small gaps between them. That said, our model consistently improves over all baselines across all datasets, highlighting an advantage even under these constrained conditions.
>
> As proof of this, in concept-complete settings such as the synthetic $CUB_c$ dataset, where we explicitly made a few child nodes deterministically dependent on their parents, our approach yields significantly larger gains.
>
> We see our work as a first step toward integrating causal reasoning with concept-based interpretability. We believe incomplete or noisy causal structures can offer a promising direction for future research.
>
>
>
> ***Q6 - Table formatting***
>
> > Do you think it might be an idea to move two columns to the appendix to improve the readability within the main text?
>
> We included incomplete datasets because we were concerned that using only the complete datasets would not clearly convey the message we aimed to communicate regarding preserving the black-box task accuracy.
> We are considering removing cMNIST, as it is a very simple task, in order to improve readability.
>
>
> ***Q7 - Citation recommendation***
>
> > I would consider citing [1] in terms of concept interventions
>
> We thank the reviewer for the suggestion, we think it is relevant to highlight the relevance and timeliness of interventions. We have added the reference at l.220.
>
>
> We hope to have clarified all of the reviewer’s concerns and are happy to provide further details if needed.
>
>
> **Refs:**
> [1] Koh et al., Concept bottleneck models, ICML, 2020

---

> > ### Comment · Reviewer_mU7E · 2025-08-05
> >
> > Thanks for the detailed response. My questions have been clarified. I remain at my score.

---

> > > ### Author Response · Authors · 2025-08-05
> > > **Feedback on rebuttal**
> > >
> > > We sincerely thank you again for supporting our work.
> > >
> > > Could you share any insights on our responses to your comments and those of the other reviewers (e.g., new ablations and sensitivity study)? Our goal is to use your feedback to make the paper as strong as it can be.

---

### Official Review · Reviewer_NX3e · 2025-07-02

**Clarity:** 3
**Significance:** 2
**Originality:** 2
**Rating:** 4
**Confidence:** 2

**Summary:**

The paper introduces Causally-reliable Concept Bottleneck Models (C2BMs), an extension of CBMs that force predictions to flow through an explicit causal graph model. Specifically, the authors propose a neural architecture that couples a concept encoder with a meta-model that adaptively parameterises linear structural equations. To build the causal graph, the authors propose an automated pipeline that discover concepts and edges using a retrieval-augmented LLM. They conduct experiments on multiple datasets showing comparable performance to other CBM architectures but find that C2BMs decrease reliance on spurious correlations and enable interventions to improve fairness.

**Questions:**

- Have you quantified the extend to which the quality of the causal graph is a determining factor for model performance?

**Ethical Concerns:**

["NO or VERY MINOR ethics concerns only"]

**Final Justification:**

The paper demonstrates how injecting domain‐level causal structure into concept bottleneck models can preserve accuracy while enabling meaningful intervention. The author rebuttal mostly addresses my main concern that a misspecified or corrupted graph might degrade performance more than in graph-agnostic baselines.

**Limitations:**

See above

**Quality:**

3

**Strengths And Weaknesses:**

**Strenghts:**
- The integration of known causal relationships into CBMs is well motivated
- The proposed architecture combining a learnable encoder with adaptive structural equations is elegant
- The evaluations demonstrates comparable performance to other CBMs while providing better steering ability
- The causal graph construction pipeline using LLMs can reduce reliance on human experts

**Weaknesses:**
- The causal model injects the graphs's inductive biases into the learner. Thus, if the graph is misspecified, these biases can hurt task accuracy relative to graph-agnostic models.
- The paper only contrasts the learned graphs against a ground truth graph but does not report performance when the model is trained with random or adversarially corrupted graphs. Thus, it's unclear how sensitive C2BM is to the quality of the causal graph.

---

> ### Author Rebuttal · Authors · 2025-07-30
>
> We thank the reviewer for their comments and feedback. As for the raised concerns and questions, individual points are addressed below.
>
> ***W1 - Performance under adversarial graph corruptions***
>
> > if the graph is misspecified, these biases can hurt task accuracy relative to graph-agnostic models.
>
> **We disagree**. C$^2$BM is built upon a key theoretical result (see Appendix D.1), which shows that it is a **universal approximator** for the final task, **regardless** of the specific causal graph. This implies that C$^2$BM retains task accuracy of black-box models, even with a largely corrupted graph, as long as the architecture is sufficiently expressive.
> Recall that, while task accuracy is preserved, causal reliability does degrade under graph corruption. This would affect the quality of explanations and out-of-distribution (OOD) performances, as shown by the differences between standard CBMs (i.e., flat, unreliable causal structures) and C$^2$BMs in Tab. 2 and Fig. 5, respectively.
>
>
> ***W2,Q1 - Sensitivity to graph corruptions***
>
> > it's unclear how sensitive C$^2$BM is to the quality of the causal graph. ... Have you quantified the extend to which the quality of the causal graph is a determining factor for model performance?
>
> Despite the theoretical argument, we agree that it is important to empirically validate C$^2$BM's robustness in task accuracy. To this end, we added a **new sensitivity analysis** evaluating task accuracy when training C$^2$BM on increasingly more corrupted graphs (3 seeds, uncertainties represent 2 sample mean $\sigma$). We consider two strategies, both starting from the ground-truth graph. In the first, we alter a percentage $p$ of graph edges by applying one of the following operations (chosen randomly, per edge): flipping, addition, or removal.
>
> | Dataset / p    | 0.05          | 0.1           | 0.2           | 0.4           | 0.6           | 0.8           | 1.0           |
> |-------------|---------------|---------------|---------------|---------------|---------------|---------------|---------------|
> | **asia**     | 71.8 ± 0.6  | 71.6 ± 0.5  | 71.43 ± 0.6  | 71.7 ± 1.3  | 70.2 ± 1.9  | 71.4 ± 1.2  | 70.2 ± 0.9  |
> | **sachs**    | 65.1 ± 1.9  | 65.6 ± 2.0  | 64.85 ± 2.0  | 64.6 ± 1.2  | 65.5 ± 1.5  | 65.2 ± 1.5  | 65.0 ± 2.0  |
> | **insurance**| 67.2 ± 1.6  | 67.0 ± 3.1  | 67.45 ± 2.3  | 66.1 ± 1.1  | 67.0 ± 2.6  | 66.8 ± 2.6  | 67.2 ± 3.1  |
> | **alarm**    | 61.9 ± 2.6  | 61.7 ± 2.1  | 61.48 ± 2.5  | 61.9 ± 2.7  | 60.6 ± 1.9  | 60.9 ± 1.6  | 61.4 ± 1.7  |
> | **hailfinder**| 74.0 ± 1.5 | 73.1 ± 3.0  | 72.32 ± 2.2  | 72.3 ± 3.2  | 73.0 ± 2.1  | 71.8 ± 1.6  | 72.4 ± 1.4  |
>
> In the second one, we progressively transform the graph into the flat structure assumed by standard CBMs, i.e., progressively connecting a percentage $p$ of nodes to the chosen task and eliminating other outgoing edges.
>
>
> | Dataset / p    | 0.05          | 0.1           | 0.2           | 0.4           | 0.6           | 0.8           | 1.0           |
> |----------------|---------------|---------------|---------------|---------------|---------------|---------------|---------------|
> | **asia**       | 72.0 ± 1.0  | 72.0 ± 1.0  | 71.5 ± 0.3  | 71.7 ± 0.8  | 71.7 ± 0.5  | 71.2 ± 1.6  | 71.1 ± 1.1  |
> | **sachs**      | 65.5 ± 2.2  | 64.8 ± 3.0  | 65.8 ± 1.6  | 64.7 ± 1.5  | 65.2 ± 1.2  | 65.1 ± 2.7  | 64.9 ± 2.2  |
> | **insurance**  | 67.4 ± 2.6  | 66.2 ± 1.7  | 66.3 ± 2.5  | 65.6 ± 3.3  | 65.3 ± 2.4  | 66.9 ± 1.8  | 64.7 ± 3.1  |
> | **alarm**      | 62.6 ± 2.9  | 61.6 ± 2.6  | 61.6 ± 2.6  | 60.5 ± 2.6  | 61.2 ± 1.8  | 60.6 ± 3.2  | 61.3 ± 1.9  |
> | **hailfinder** | 73.1 ± 1.1  | 73.2 ± 1.9  | 72.5 ± 0.9  | 72.7 ± 2.9  | 73.0 ± 2.6  | 73.0 ± 1.7  | 72.9 ± 1.9  |
>
> The results confirm that task accuracy **remains stable** even when the causal graph is heavily corrupted, empirically supporting the theoretical claim. We thank the reviewer for raising this point, we will extend the paper with this new experiment.
>
> We hope to have clarified all of the reviewer’s concerns and are happy to provide further details if needed.

---

> > ### Comment · Reviewer_NX3e · 2025-08-05
> >
> > Thank you for the clarification and additional experiments. This addresses my main concern and I will increase my score to 4 accordingly.

---

> > > ### Author Response · Authors · 2025-08-06
> > >
> > > We sincerely thank you again for supporting our work.
> > >
> > > We would be happy to incorporate any further feedback on our response to your comments and those of the other reviewers.

---

### Official Review · Reviewer_6yXN · 2025-07-03

**Clarity:** 3
**Significance:** 3
**Originality:** 3
**Rating:** 5
**Confidence:** 3

**Summary:**

This work deals with the task of building interpretable predictive models for raw data like image and text. While existing methods like Concept Bottleneck Models (CBMs) are transparent in their decision-making, their explanations are not causal, that is, they cannot be used to recommend interventions. This work extends CBMs by making them causal, calling the result Causally reliable Concept Bottleneck Models (C2BMs). This model consists of a set of concepts, a structural causal model (SCM) over the concepts, a neural encoder mapping raw data to a set of latent representations, and a neural encoder mapping each latent representation to a set of weights, used to define structural equations of the SCM.
The authors do so using concept bottleneck, that is, using a neural network to map the raw data to a set of interpretable concepts, which are used to make the final decision. The authors extend CBMs by combining them with causal models, learning neural encoders to map raw data to structural causal models of concepts, allowing reliable causal inference.
The proposed method consists of three stages. The first stage is concept discovery; they use label free concept discovery and then filter the concepts based on brevity, distinctness and presence in their training data. A pretrained vison-language model like CLIP is used to annotate each data point with these concepts. The second stage is causal discovery; they address this by first constructing a CPDAG using greedy equivalence search and then orienting its edges by querying an LLM. To improve reliability, retrieval augmented generation (RAG) and majority voting across 10 repetitions is used. Finally, the third stage involves learning the structural equations; they approximate each such equation for each variable as a weighted sum of its parents’ values, each parent weighed by the neural encoder’s output. Each structural equation is learned separately, exploiting the Markov condition due to the SCM’s causal graph.
The authors evaluate this approach on 5 BN-based domains, 1 benchmark domain color MNIST (cMNIST) domain, and 3 real-world domains: CelebA, facial recognition, annotated with binary facial attributes; CUB_c, a subset of original CUB birds data set, having a subset of concepts and some new ones to introduce deeper causal relationships; and Siim-Pneumothorax, consisting of chest radiograms, annotated with a single label indicating presence of Pneumothorax, augmented with annotations extracted by label-free methods. The empirical evaluation shows that (1) the causal discovery method accurately recovers the underlying causal relationships and C2BM performs on par or better than baseline models, including an opaque NN, (2) C2BM achieves better performance with fewer interventions, (3) causal structure allows C2BM to mitigate the effect of spurious correlations captured by neural encoder, (4) causal structure allows C2BM to eliminate unfair bias from its decision-making.

**Questions:**

Questions
1. What is the effect of data set size and causal graph complexity on the causal discovery method?
2. While the RAG context has no impact in the Alarm domain, it improves performance on the Sachs domain. Were there any cases where using RAG degraded performance?

**Ethical Concerns:**

["NO or VERY MINOR ethics concerns only"]

**Final Justification:**

The authors have addressed most of the concerns. So, I have increased my score to 5 (accept)

**Limitations:**

yes

**Quality:**

3

**Strengths And Weaknesses:**

Strengths
1. Highly relevant problem for high stakes domains like healthcare that require interpretability and causal reliability
2. The proposed pipeline is general enough to use more specialized LLM inference frameworks e.g., Loula et al., (2025).

Weaknesses
1. Since LLMs are stochastic and sensitive to phrasing of the prompt, the design of the prompt might affect performance.
2. Since LLMs answer causal questions by approximately retrieving causal facts from their training data, causal graph construction would be harder in understudied domains like rare diseases.

References
Loula, João, et al. "Syntactic and Semantic Control of Large Language Models via Sequential Monte Carlo." ICLR, 2025.

---

> ### Author Rebuttal · Authors · 2025-07-30
>
> We thank the reviewer for their comments and feedback. As for the raised concerns and questions, individual points are addressed below.
>
> As a side note, we found that the citation mentioned by the reviewer could support our Remark 4.1. We have decided to include it; thank you for the pointer.
>
> ***W1 - Prompt design***
>
> > the design of the prompt might affect performance.
>
> **We agree**. The design of the prompt could condition the causal graph refinement. To assess this, we have added a **new ablation** study in which we fix the incomplete causal graph generated by the causal discovery algorithm, and later evaluate variants of the LLM+RAG refinement. Specifically, we evaluate the impact of four different prompting strategies with two LLMs (GPT-4o, $\sim$200M parameters; and GPT-4o-mini, $\sim$8M parameters). The considered strategies are as follows:
> - *minimal* prompting: simply asks the LLM to identify causal relations without additional guidance;
> - *instruction* prompting: provides a more detailed explanation of what constitutes a causal relation;
> - *few-shot* prompting: proposed in [1], combines instruction prompting with a few examples;
> - *Chain-of-Thought (CoT)*: proposed in [2], encourages the model to generate intermediate logical steps before generating the answer. This is the approach we used in the original manuscript.
>
> The table below shows the number of Mistaken Edges for datasets with known ground-truth causal graphs (reliability of standard, flat, CBMs is also reported for reference).
>
> | LLM         | Prompting Strategy | Asia | Sachs | Insurance | Alarm | Hailfinder |
> | ----------- | ------------------ | ---- | ----- | --------- | ----- | ---------- |
> | flat CBMs   |         --         | 11   | 23    | 74        | 78    |     117    |
> | gpt-4o      | minimal            | 1    | 12    | 19        | 11    |     22     |
> | gpt-4o      | instruction        | 1    | 11    | 19        | 11    |     22     |
> | gpt-4o      | few\_shot          | 1    | 8     | 19        | 11    |     22     |
> | gpt-4o      | CoT (paper)        | 1    | 7     | 18        | 9     |     22     |
> | gpt-4o-mini | minimal            | 1    | 10    | 19        | 10    |     22     |
> | gpt-4o-mini | instruction        | 1    | 9     | 19        | 11    |     22     |
> | gpt-4o-mini | few\_shot          | 1    | 10    | 19        | 11    |     22     |
> | gpt-4o-mini | CoT                | 1    | 7     | 19        | 10    |     22     |
>
> Despite moderate variation in the Sachs dataset, the performance is generally very **robust** to the prompt strategy, and causal reliability is **largely superior to standard CBMs** in all cases.
>
>
>
> ***W2 - Understudied domains***
>
> > causal graph construction would be harder in understudied domains like rare diseases
>
> This is undoubtedly true, as we mentioned in our limitations (l.366). However, we believe understudied domains leave very little room for any alternative method. Note that, in any case, our pipeline can still approximate causal relations from observational data only. We hence believe C$^2$BM could be useful for exploratory research in understudied domains to restrict the search space of possible hypotheses and contribute to forming a background knowledge.
>
>
>
> ***Q1 - Sample complexity of graph discovery***
>
> >  What is the effect of data set size and causal graph complexity on the causal discovery method?
>
> We explored this with a **new sensitivity study**, running the causal graph pipeline (causal discovery + LLM refinement with CoT prompt) across all datasets with an available ground-truth graph. For each, we varied the number of data points $N$ from 100 to 10,000.
>
> | Dataset    | Flat CBMs |                |      |   C$^2$BM's   |   causal     |  graph    |                 |
> |------------|-----------|------------------------|------|------|------|------|-----------------|
> |            |           | **Data size** $\rightarrow$          | **100**  | **500**  | **1000** | **5000** | **10000 (Paper)**   |
> | **Asia**       | 11        |                        | 6    | 3    | 1    | 1    | 1               |
> | **Sachs**      | 23        |                        | 15   | 12   | 11   | 7    | 7               |
> | **Insurance**  | 74        |                        | 47   | 35   | 24   | 22   | 18              |
> | **Alarm**      | 78        |                        | 31   | 16   | 9    | 9    | 9               |
> | **Hailfinder** | 117       |                        | 56   | 47   | 44   | 22   | 22              |
>
> A few considerations:
>
> - C$^2$BM's causal graph is consistently more causally reliable than the flat structure implicitly assumed by standard CBMs, regardless of the dataset size.
> - As expected, increasing the number of data points leads to better alignment between the estimated and true causal graphs. Causal reliability tends to remain stable at larger data sizes.
> - We observed that the data size threshold for reliable performance does not strictly depend on graph size. We speculate this is due to the varying impact of LLM-based refinement across datasets. In some cases, an effective background knowledge can compensate for limited data.
>
> We appreciate the reviewer’s question and will incorporate these findings into the paper.
>
>
> ***Q2 - Clarification on RAG performance***
>
> > Were there any cases where using RAG degraded performance?
>
> No, we did not observe any cases where using RAG degraded performance. We speculate this is because all our investigated domains are well-supported by archived scientific literature. In the paper, we included results for all the datasets we experimented on.
>
>
> We hope to have clarified all of the reviewer’s concerns and are happy to provide further details if needed.
>
> **Refs:**
> [1] Ye et al., Investigating the effectiveness of task-agnostic prefix prompt for instruction following, AAAI, 2024
> [2] Wei et al., Chain-of-thought prompting elicits reasoning in large language models, NeurIPS, 2022

---

> > ### Comment · Area_Chair_mKTJ · 2025-08-07
> >
> > Dear Reviewer mKTJ,
> >
> > A requirement for reviewing is to engage in discussion with the reviewers; rebuttals to your review. The authors have provided a thoughtful rebuttal to your review. Please be explicit in stating if their responses have addressed your concerns and merit reevaluating their work, or if there are still hurdles to you improving your rating and what they are.
> >
> > Kind Regards,
> >
> > AC

---

### Official Review · Reviewer_wNgz · 2025-07-05

**Clarity:** 2
**Significance:** 2
**Originality:** 2
**Rating:** 2
**Confidence:** 4

**Summary:**

This paper proposes Causally reliable Concept Bottleneck Models (C2BMs), which explicitly incorporate causal structure into the concept bottleneck architecture. The authors present a novel pipeline to automatically learn the causal concept structure from observational data and unstructured knowledge sources, aiming to enhance interpretability, causal reliability, and responsiveness to interventions.

**Questions:**

1. The notion of a "bottleneck" presented in the paper is intriguing. However, the manuscript lacks details on how the bottleneck is quantitatively evaluated and how this mechanism influences the model’s operations. I recommend the authors provide a more thorough explanation and possibly include experimental evidence demonstrating its effect.

2. Section 4 cites and utilizes many existing methods, which may obscure the unique contributions of the current study. To improve clarity, the authors should explicitly state the advantages of their approach and clearly differentiate it from prior works.

3. The proposed model does not appear to achieve significantly better task accuracy compared to baseline methods. The authors should discuss possible reasons for this and highlight the practical or theoretical benefits that justify their approach despite the modest performance gains.

4. The training procedure and other experimental details are insufficiently described. Please provide a comprehensive explanation of how the model is trained. Additionally, we suggest including ablation studies from various perspectives discussed in Section 4, such as concept discovery, labeling, and other relevant components, to better understand the contribution of each part.
5. As the authors mentioned, a robust prior knowledge base is very important to model training and learning. This part is interesting, but not including many details. Suggest the authors explain how to built such a knowledge base. and answer how to evaluate the effectiveness of the built knowledge base, cause that can impact the training performance. Is the pipeline for constructing the knowledge base fully automated? How is the transferability of the knowledge ensured?

**Ethical Concerns:**

["NO or VERY MINOR ethics concerns only"]

**Final Justification:**

I have carefully read the author's response and, after rigorous evaluation, I believe that this paper does not meet the acceptance level of NISP. I retain my previous rating.

**Limitations:**

yes

**Quality:**

2

**Strengths And Weaknesses:**

The research tackles an important challenge at the intersection of explainability and causality in deep learning and offers promising directions. However, further clarification on the automatic structure learning process and more extensive evaluation would strengthen the manuscript.

---

> ### Author Rebuttal · Authors · 2025-07-30
>
> We thank the reviewer for their comments and feedback. As for the raised concerns and questions, individual points are addressed below.
>
> ***Q1 - Bottleneck evaluation***
> > the manuscript lacks details on how the bottleneck is quantitatively evaluated and how this mechanism influences the model’s operations.
>
> The standards of evaluation of concept bottlenecks [1] are well established in the CBM literature [1,2,3,4,5,6]. These are evaluated in terms of **interpretability** (e.g., concept accuracy) and **actionability** (e.g., interventions) (ll.19-26).
> Matching task accuracy of black-box models is also an important desideratum (ll.70-72), yet not trivial when conditioned to provide human-interpretable and actionable models. In fact, this often exceeds typical expectations of the CBM community [7].
>
>
> Regarding our proposed *causal* concept bottleneck, **we disagree that a quantitative evaluation is lacking** in our manuscript. Section 5 provides extensive evaluations of our causal bottleneck, assessing standard CBMs' desiderata (see above) while also evaluating further specific benefits of our model.
>
> Specifically:
> - (Task acc.) **Tab. 1** shows that our causal bottleneck preserves the task accuracy of black-box models, in line with CBMs' desiderata.
> - (Concept acc.) **Tab. 6** shows that our causal bottleneck preserves the concept accuracy of prior CBMs.
> - (Interpretability) **Tab. 2** shows our bottleneck provides interpretability at a deeper level by aligning more closely with ground-truth causal structures than prior CBMs.
> - (Actionability) **Fig. 4** shows how our bottleneck permits more effective interventions than prior CBMs. **Fig. 5** and **Tab. 3** evaluate two C$^2$BM's critical benefits: permit human interventions to de-bias the model from spurious correlations and unfair behavior.
>
> We hope this clarifies that our experimental evaluation is centered on the causal bottleneck and its contributions.
>
>
> ***Q2 - Clarify distinction with prior works***
> > Section 4 cites and utilizes many existing methods, which may obscure the unique contributions of the current study.
>
> Thank you for your constructive comment, as it allows us to improve clarity.
>
> Our core contribution is the C$^2$BM model (right side of Fig. 2), a novel extension of CBMs that, instead of assuming a flat layer of independent concepts,  forces inference to flow through a **given** causal graph of concepts (ll.73-76). As stated in the paper (ll.42-44, ll.140-141, Remark 4.1 in ll.150-152), C$^2$BM requires **as input** a set of concepts (as most CBMs) and a causal graph connecting them; these are **distinct** elements that must be provided prior to C$^2$BM training.
>
> However, these requirements are often not readily available in real-world scenarios (ll.140-142). We argue that fully automating the instantiation of C$^2$BM is both valuable and non-trivial. For this reason, we present a pipeline that enables full automation (ll.142-145) as a second contribution of our work; this justifies its placement in the Method section. While the individual components of the pipeline are mostly based on prior work (hence the citations), their integration into a coherent, automated process -- which was never published before -- is novel by definition.
>
> To improve clarity, we have added a clarification in the main text after l.149: "In the following, we outline our implementation for each sub-problem. Specifically, building C$^2$BM's individual prerequisites will be mostly based on prior work. Note that integrating them into a coherent, automated pipeline is instead part of this paper's contributions.". We thank the reviewer for raising this point and hope this resolves the concern.
>
>
> > the authors should explicitly state the advantages of their approach and clearly differentiate it from prior works.
>
> C$^2$BM is a kind of CBM architecture, **not** a method for concept or causal discovery, hence these do not constitute direct prior work from which C$^2$BM needs to be differentiated. In the paper, we explicitly state our advantages in terms of actionability and causal reliability w.r.t. prior methods in the CBMs community: in the introduction (ll.27-37), preliminaries (ll.73-76), related works (ll.110-112), and conclusions (ll.351-355).
>
>
> ***Q3 - Task accuracy***
> > The proposed model does not appear to achieve significantly better task accuracy compared to baseline methods.
>
> **We firmly disagree that this should be seen as a weakness.** The ultimate goal of our field of research is to improve on interpretability and actionability aspects while **preserving** the task accuracy of traditional black-box approaches [2,6]. As demonstrated in Tab.1, C$^2$BM matches the performance of the corresponding black-box architecture, which is considered an upper limit in the interpretability community and generally difficult to achieve [7]. Furthermore, we support this result with a proof stating C$^2$BM is a universal approximator for the downstream task (see Appendix D.1). Therefore, the rest of the experimental analysis focuses on aspects critical to the interpretability community, i.e., reliability, actionability, and fairness.
>
>
> ***Q4.1 - Training details***
> > The training procedure and other experimental details are insufficiently described.
>
> We disagree. C$^2$BM's training objective is formally described in **Eq.6**, i.e., maximum likelihood (similar to other CBM classifiers [2]) conditioned on endogenous parents' values and meta-model output. Full experimental details, including numerical values for training hyperparameters, are provided in **Appendix~F** (referenced in ll.243-244). Finally, for the most comprehensive list, we provide the **code** with all settings specified in the YAML configuration files.
>
>
> ***Q4.2 - Ablations***
> > we suggest including ablation studies from various perspectives ... such as concept discovery, labeling, and other relevant components
>
> We value the reviewer's suggestion. A few ablations were already included in Appendix G due to space constraints. Specifically: (i) **Appendix G.4** provides an extensive ablation on the causal discovery block. (ii) **Appendices G.5–G.6** analyze components related to querying the LLM. In addition, we have now included: (iii) a new ablation on the LLM prompt (please refer to our response **W1 to Reviewer 6yXN**, and (iv) a study on robustness to corrupted causal graphs (response **W2 to Reviewer NX3e**).
>
>
>
> ***Q5 - Background knowledge***
> > explain how to build such a knowledge base. ... Is the pipeline for constructing the knowledge base fully automated?
>
> There might be a misunderstanding here. **We do not build the knowledge base ourselves**. Instead, as in ll.180-181 and detailed in Appendix C.2, we leverage a **pretrained LLM**, which implicitly encodes broad, **unstructured** knowledge acquired during its training phase [3]. We further improve this with a RAG strategy, which allows the model to take archived domain-specific knowledge (e.g., scientific papers), available through public sources such as *arXiv*, automatically into account (ll.97-99 and Appendix C.2).
>
>
> > how to evaluate the effectiveness of the built knowledge base
>
> We evaluate its effectiveness by measuring how much it improves the alignment between the learned graph and the ground-truth graph (see '+LLM' row in **Tab. 2**). This structure constrains C$^2$BM reasoning, making predictions and explanations intrinsically more aligned with real-world causal mechanisms.
>
>
> > How is the transferability of the knowledge ensured?
>
> Transferability of the knowledge base to the model is ensured **by construction**: each concept value is predicted from its causal parents as defined in the graph refined with the knowledge base. This forces a principled flow of information aligned with expert knowledge. Nevertheless, we are not sure we fully understood the reviewer's question. We kindly ask the reviewer to reformulate should our response be off-target.
>
>
> We hope our responses offer a clearer perspective on the paper and help position its contribution within the interpretability community. We are happy to further engage in the discussion.
>
> **Refs:**
> [1] Koh et al., Concept bottleneck models, ICML, 2020
> [2] Zarlenga et al., Concept embedding models, NeurIPS, 2022
> [3] Oikarinen et al., Label-Free Concept Bottleneck Models, ICLR, 2023
> [4] Zarlenga et al., Learning to Receive Help: Intervention-Aware Concept Embedding Models, NeurIPS, 2023
> [5] Poeta et al., Concept-based explainable artificial intelligence: A survey, Arxiv, 2023
> [6] Ismail et al., Concept Bottleneck Generative Model, ICLR, 2024
> [7] Arrieta et al., Explainable Artificial Intelligence (XAI): Concepts, taxonomies, opportunities and challenges toward responsible AI. Information fusion, 2020

---

> > ### Comment · Reviewer_wNgz · 2025-08-05
> > **Thanks for the response.**
> >
> > Thanks for the response. I don't have much questions.

---

> > > ### Author Response · Authors · 2025-08-05
> > > **Further feedback on pending discussion items**
> > >
> > > Thank you for answering.
> > >
> > > However, from the rebuttal, it emerged that we are not aligned on key discussion items about our paper. Specifically, pending points are the following:
> > > - **Q1** - bottleneck evaluation
> > > - **Q3** - task accuracy
> > > - **Q4.1** - training details
> > > - **Q4.2** - additional ablations (**W1** to **Reviewer 6yXN**, **W2** to **Reviewer NX3e**) and sensitivity studies (**Q1** to **Reviewer 6yXN**)
> > > - **Q5** - background knowledge
> > >
> > > Could you share your targeted feedback on each point? They are important for us to fully understand your revised perspective on the paper.

---

> > > > ### Comment · Area_Chair_mKTJ · 2025-08-07
> > > > **Please continue discussion with authors**
> > > >
> > > > Dear Reviewer wNgz,
> > > >
> > > > A requirement for reviewing is to engage in discussion with the reviewers; rebuttals to your review. The authors have provided a thoughtful rebuttal to your review. Please be explicit in stating if their responses have addressed your concerns and merit reevaluating their work, or if there are still hurdles to you improving your rating and what they are.
> > > >
> > > > Kind Regards,
> > > >
> > > > AC

---

### Comment · Area_Chair_mKTJ · 2025-08-05
**Reminder to respond to authors' rebuttals**

Hi Reviewers,

Please kindly follow up on the authors' rebuttal to your review, as we are approaching the end of the discussion period.

Thanks.

Regards,

AC

---

### Author Response · Authors · 2025-08-08
**Additional results with GPT-5**

Dear reviewers,

Given today’s release of GPT-5, we thought it would be valuable to add new results using this new LLM in the graph-refinement step of our pipeline.

The table below represents an extension of the table we presented in our response **W1** to **Reviewer 6yXN**. It shows the number of Mistaken Edges for datasets with known ground-truth causal graphs (reliability of standard, flat, CBMs is also reported for reference).

| LLM         | Prompting Strategy | Asia | Sachs | Insurance | Alarm | Hailfinder |
| ----------- | ------------------ | ---- | ----- | --------- | ----- | ---------- |
| flat CBMs   |         --         | 11   | 23    | 74        | 78    |     117    |
| gpt-4o      | minimal            | 1    | 12    | 19        | 11    |     22     |
| gpt-4o      | instruction        | 1    | 11    | 19        | 11    |     22     |
| gpt-4o      | few\_shot          | 1    | 8     | 19        | 11    |     22     |
| gpt-4o      | CoT (paper)        | 1    | 7     | 18        | 9     |     22     |
| gpt-4o-mini | minimal            | 1    | 10    | 19        | 10    |     22     |
| gpt-4o-mini | instruction        | 1    | 9     | 19        | 11    |     22     |
| gpt-4o-mini | few\_shot          | 1    | 10    | 19        | 11    |     22     |
| gpt-4o-mini | CoT                | 1    | 7     | 19        | 10    |     22     |
| **gpt-5 (new)**       | minimal            | 1    | 7     | 18        | 10    |     22     |
| **gpt-5 (new)**       | instruction        | 1    | 2     | 17        | 10    |     22     |
| **gpt-5 (new)**       | few\_shot          | 1    | 2     | 16        | 10    |     22     |
| **gpt-5 (new)**       | CoT                | 1    | 3     | 16        | 10    |     22     |

The results show that:
- Using GPT-5, **causal reliability improves** on average w.r.t. the configuration we report in the original paper (gpt-4o, CoT), especially on the *Sachs* dataset.
- With this, we also wanted to showcase how our pipeline can **readily incorporate novel approaches** for knowledge querying (ll.150-151), as also endorsed by Reviewer 6yXN.
- Using GPT-5, performance is generally **robust** to the prompt strategy (made an exception for naive *minimal* strategy).

We hope this new addition helps support the flexibility of the pipeline.

The authors

---

### Decision · Program_Chairs · 2025-09-17

**Decision:**

Accept (poster)

**Comment:**

This paper extends concept bottleneck models (CBMs) by introducing a causal structure over concepts, producing causally reliable CBMs (C2BMs). A pipeline is proposed to automatically learn such structures from observational data and background knowledge via LLMs. Experiments suggest that C2BMs improve interpretability, causal reliability, and intervention responsiveness while maintaining accuracy.

**Strengths**

* Tackles an important and timely problem at the intersection of interpretability and causality.

* Elegant architecture combining concept encoders with structural causal equations.

* Pipeline to integrate observational data and LLM-based causal discovery is novel and flexible.

* Demonstrates causal reliability, intervention capacity, and fairness benefits compared to standard CBMs.

Authors provided extensive rebuttal clarifications, new ablations, sensitivity studies, and updated results (including with GPT-5).

**Weaknesses**

* Initial clarity issues: some reviewers found the description of bottleneck evaluation, training details, and role of causal discovery confusing.

* Concerns about empirical scope and sensitivity to graph misspecification, though rebuttal added robustness experiments addressing this.

* Improvements in accuracy over baselines are modest; value lies primarily in reliability and interpretability.

The consensus among engaged reviewers leans toward acceptance, with clear appreciation of the contributions despite some limitations.

**Recommendation**
This work makes a solid contribution by bridging CBMs with causal modeling and proposing an automated, LLM-based pipeline for structure discovery. While empirical gains in accuracy are limited, the advances in interpretability, reliability, and intervention justify acceptance.

Final recommendation: Accept.